# Genome-wide association study of thyroid-stimulating hormone highlights new genes, pathways and associations with thyroid disease

Alexander T. Williams[1,19] ✉, Jing Chen[1,19], Kayesha Coley[1], Chiara Batini[1,2], Abril Izquierdo[1,2], Richard Packer[1,2], Erik Abner[3], Stavroula Kanoni[4], David J. Shepherd[1], Robert C. Free[2,5], Edward J. Hollox[6], Nigel J. Brunskill[7], Ioanna Ntalla[1], Nicola Reeve[1,8], Christopher E. Brightling[2,9], Laura Venn[1], Emma Adams[1], Catherine Bee[1], Susan E. Wallace[1], Manish Pareek[2,10], Anna L. Hansell[1], Tõnu Esko[3], Estonian Biobank Research Team*, Daniel Stow[11], Benjamin M. Jacobs[12,13], David A. van Heel[14], Genes & Health Research Team*, William Hennah[15,16,17], Balasubramanya S. Rao[18], Frank Dudbridge[1], Louise V. Wain[1,2], Nick Shrine[1], Martin D. Tobin[1,2,20] & Catherine John[1,2,20] ✉

Thyroid hormones play a critical role in regulation of multiple physiological functions and thyroid dysfunction is associated with substantial morbidity. Here, we use electronic health records to undertake a genome-wide association study of thyroid-stimulating hormone (TSH) levels, with a total sample size of 247,107. We identify 158 novel genetic associations, more than doubling the number of known associations with TSH, and implicate 112 putative causal genes, of which 76 are not previously implicated. A polygenic score for TSH is associated with TSH levels in African, South Asian, East Asian, Middle Eastern and admixed American ancestries, and associated with hypothyroidism and other thyroid disease in South Asians. In Europeans, the TSH polygenic score is associated with thyroid disease, including thyroid cancer and age-of-onset of hypothyroidism and hyperthyroidism. We develop pathway-specific genetic risk scores for TSH levels and use these in phenome-wide association studies to identify potential consequences of pathway perturbation. Together, these findings demonstrate the potential utility of genetic associations to inform future therapeutics and risk prediction for thyroid diseases.

Thyroid hormones are essential for energy metabolism and act on almost all cells. Thyroid dysfunction is associated with secondary cardiovascular, mental health, ophthalmic and other disease[1]. Hypothyroidism has a high prevalence[2] and is most commonly due to autoimmune (Hashimoto) thyroiditis, in areas where iodine intake is sufficient[1]. Hyperthyroidism, prevalence 0.2–1.3%, is most commonly due to autoimmune (Graves) disease or toxic nodular goitre[1]. Ageing, diet (including iodine deficiency), smoking status, genetic

---

A full list of affiliations appears at the end of the paper. *Lists of authors and their affiliations appear at the end of the paper. ✉e-mail: atw20@leicester.ac.uk; cj153@leicester.ac.uk

susceptibility, ethnicity, and endocrine disruptors are risk factors for thyroid diseases; defining genetic variants, genes, proteins and pathways associated with hypothyroidism and hyperthyroidism will inform a deeper understanding of the mechanisms of thyroid disease and inform prevention and treatment strategies.

Genome-wide association studies (GWAS) of quantitative traits have been particularly powerful and successful in identifying new drug targets[3–5]. Most genetic associations for thyroid disorders were discovered in GWAS of thyroid-stimulating hormone (TSH) levels, a sensitive marker of thyroid function which is suppressed when levels of thyroxine ($T_4$) and triiodothyronine ($T_3$) are high and elevated when $T_4$ and $T_3$ levels are low. The largest GWAS to date, including 119,715 participants, brought the number of known genetic associations for TSH to 99, highlighting associations with hypothyroidism, hyperthyroidism, and thyroid cancer[6].

Electronic health records (EHR) are increasingly utilised in genomic studies[7,8]. In the UK, primary care EHR have been recorded prospectively for more than 25 years. TSH is frequently measured in primary care because thyroid disease may present with non-specific symptoms or be asymptomatic. Through harnessing such TSH measures, our study included 247,107 participants, more than doubling the size of the largest study to date, increasing the number of genetic associations for TSH from 99 to 260. Using these 260 associated variants we then (i) tested the association between TSH-associated variants and disease; (ii) fine-mapped associations through annotation-informed credible sets; (iii) applied a consensus-based framework to systematically investigate and identify putative causal genes, integrating eight locus-based or similarity-based criteria; (iv) developed and applied a polygenic score (PGS) for TSH to show associations with susceptibility and age-of-onset of thyroid disease; (v) applied phenome-wide association studies (PheWAS) to individual variants, the PGS, and molecular pathway-specific genetic risk scores (GRSs). Through evidence from the above, we aimed to define putative causal genes, and provide new insights into the mechanistic pathways underlying thyroid disorders and their relationship to other long-term conditions to inform relevant drug therapies.

## Results

In UK Biobank and the EXCEED study we undertook GWAS with TSH levels, using an inverse normal transformation and adjusting for age, genotyping array, sex and the first 10 principal components of ancestry (Online Methods). Across the two studies, we analysed 127,392 European ancestry (EUR) participants. We meta-analysed summary statistics from GWAS in UK Biobank and EXCEED with those from the independent European-ancestry populations of Zhou et al[6], bringing the total sample size to 247,107 participants and 57,524,162 genetic variants in Stage 1. Sentinel variants reaching $P < 5 \times 10^{-8}$ in Stage 1 were taken forward to Stage 2, in which TSH associations were tested in 63,326 EUR participants from the Estonian Biobank[9] and 33,171 South Asian ancestry (SA) participants from Genes & Health[10]. We meta-analysed summary statistics from Stages 1 and 2. Overall, we studied TSH associations in 343,604 individuals and up to 70,647,331 genetic variants (Fig. 1).

### TSH association with 260 sentinel variants

In Stage 1 ($N = 247{,}107$), we identified 260 independent sentinel variants associated with TSH ($P < 5 \times 10^{-8}$) at 156 unique genomic loci, of which 158 sentinel variants at 78 genomic loci are new (Online Methods, Supplementary Figure 1). In addition to reaching $P < 5 \times 10^{-8}$ in the Stage 1 meta-analysis, 230 of 249 sentinel variants available in Stage 2 (133 of the 158 novel sentinel variants) reached $P < 5 \times 10^{-8}$ after meta-analysing Stages 1 and 2 ($N = 343{,}604$, Supplementary Data 2)[6,9,11].

Together the 260 sentinel variants explain 22.8% of the TSH variance (Eq. (1)), accounting for 35.1% of the heritability previously estimated by Panicker et al. at 65% (Online Methods). The median number

of variants per 95% credible set (i.e. the set of variants that has 95% probability of containing the causal variant) was 3, and 167 (64%) of credible sets had a putative causal variant with a posterior inclusion probability (PIP) > 50%. Sentinel variants were defined as the variant in each credible set with the highest posterior probability (Online Methods).

### Identification of putative causal genes and causal variants

To better understand the functional relevance of our sentinel variants, we undertook comprehensive variant-to-gene mapping by integrating evidence from eight methods: (i) the nearest gene to the sentinel variant; (ii) the gene with the highest polygenic priority score (PoPS)[12]; identification of (iii) expression quantitative trait loci (eQTL) or (iv) protein quantitative trait loci (pQTL) within the credible sets; (v) proximity to a gene for a thyroid-associated Mendelian disease (±500 kb); (vi) an annotation-informed credible set containing a missense/deleterious/damaging variant with a posterior probability of association >50%; (vii) identification of a rare variant (±500 kb of a TSH sentinel variant) association with hypo- or hyperthyroidism using whole-exome[13] and whole-genome sequencing[14] resources; and (viii) proximity to a human ortholog of a mouse knockout gene with a thyroid-related phenotype (±500 kb).

We identified 112 putative causal genes satisfying ≥2 criteria, of which 30 were supported by ≥3 criteria (Fig. 2, Supplementary Data 3). 36 of the 112 overlap with a list of 67 previously reported genes (Supplementary Data 4)[6,15] typically implicated by a single criterion.

Of the 112 putative causal genes supported by ≥2 criteria, 76 genes have not been previously implicated in TSH levels. The 36 previously reported genes were supported by additional criteria compared with the original reports, among which were 15 genes also supported by additional sentinel variants (Supplementary Data 3). Among the 30 genes supported by ≥3 criteria, 18 were not previously implicated in TSH levels (**ADCY6, ANXA5, BCAS3, BNC2, CADM1, HMGA2, KIAA1217, KRT18, PDE4D, PHC2, PTEN, SDCCAG8, SGK1, SMOC2, SPPL3, SULF1, TRIM2, TSHZ3**, novel genes shown in bold) and 12 were previously reported genes supported by additional criteria compared with the original reports, among which were 6 genes also supported by additional novel sentinel variants (*TG, TSHR, GLIS3, IGFBP5, PTPRS, SPATA13*). The 30 genes supported by ≥3 criteria include genes involved in transcriptional regulation (**BNC2, HMGA2, PHC2, TSHZ3**, *GLIS3*), production, signaling or response to thyroid hormones (*TG, TPO, TSHR,* **PDE4D**) or non-thyroid hormones (**ADCY6**, *INSR, NR3C2*), regulation of thyroid-relevant pathways (**HMGA2**, *IGFBP5*), neuronal protection and neuropathies (**ADCY6, TRIM2**), angiogenesis (**SMOC2**, **SKG1**, *VEGFC, SPATA13*), AKT signalling (**PTEN, SGK1**, *AKT1, PTPRS*) and ciliogenesis (**SDCCAG8**).

To supplement understanding of the biological pathways and clinical phenotypes influenced by TSH-associated variants, we first tested associations between our sentinel variants and circulating free $T_4$ levels, hypothyroidism, hyperthyroidism (Fig. 2, Supplementary Data 5), thyroid cancer and other thyroid disease in UK Biobank. Using DeepPheWAS v0.2.9[7], we then undertook PheWAS in UK Biobank of 64 sentinel variants which mapped to putative causal genes implicated by ≥3 criteria or by a single putative causal missense variant (PIP > 50%; Supplementary Fig. 2, Supplementary Data 6).

TSH sentinel variants implicating putative causal genes with ≥2 variant-to-gene mapping criteria show variable patterns of association with hypothyroidism, hyperthyroidism, thyroid cancer and other thyroid diseases. Among these are sentinel variants associated with hypothyroidism but not hyperthyroidism (implicating *AKT1, IGFBP5, INSR, GLIS3, SPATA13,* **CADM1, BCAS3**, *SASH1,* **PDE4D**, *VEGFC,* **SPPL3, BNC2**, *PDE10A, PDE8B,* **NR3C2**, *VAV3, SOX9, B4GALNT3, CGA, C9orf92, NEK6,* **NSF, CCBE1**, *GNG7, TPPP, WNT4,* **SNX8, C1orf116, RBM47, KCTD5, PPP2R1B, PTPRJ**, *OCLN,* **C9orf156, GATA3, WWTR1**, *MAL2,* **ZBTB17, ARNT, CDC16**), sentinel variants associated with

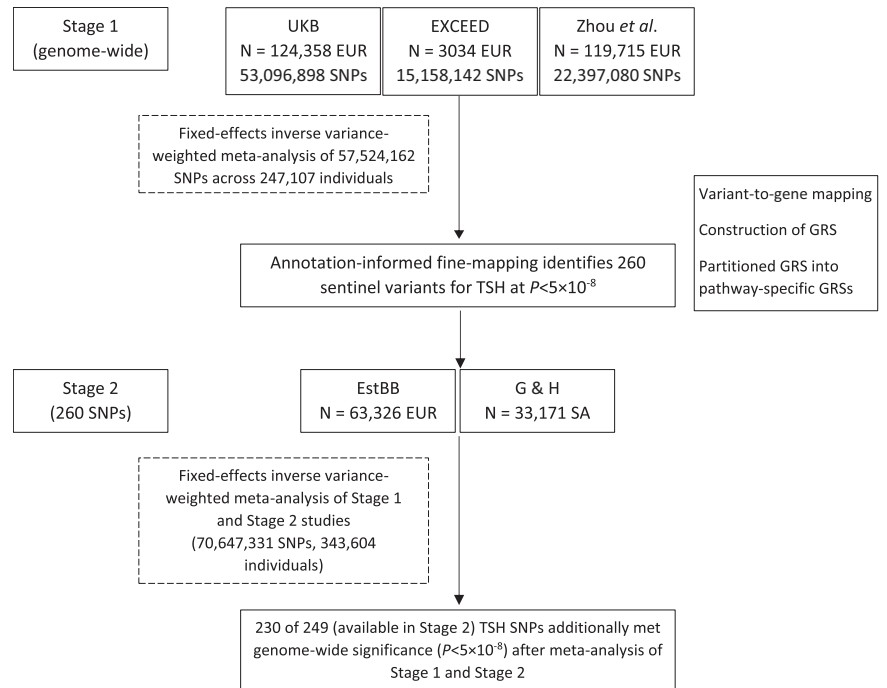

**Fig. 1 | Overview of the study design.** Flow diagram summarising the two-stage study design. UKB UK Biobank, EUR European ancestry, SA South Asian ancestry, SNP single nucleotide polymorphism, GRS genetic risk score, EstBB Estonian Biobank, G&H Genes & Health.

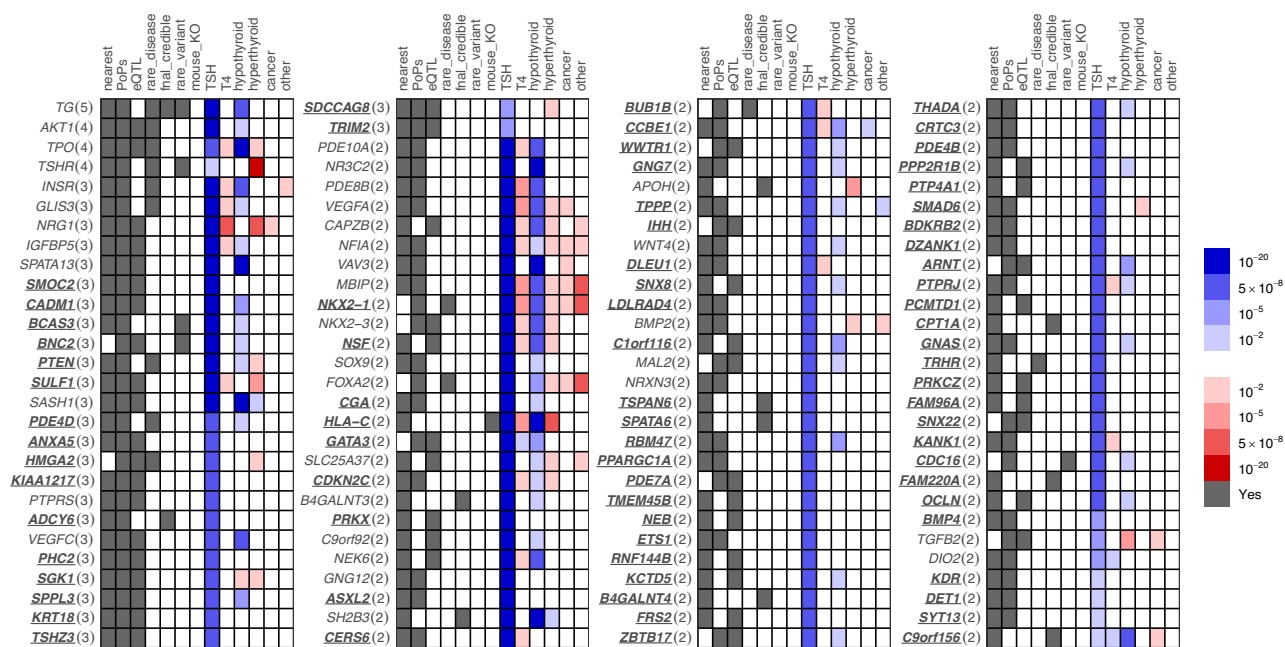

**Fig. 2 | 112 genes prioritised by two or more variant-to-gene criteria.** The first seven columns indicate that at least one variant implicates the corresponding gene via the evidence for that column (Supplementary Data 3). The remaining six columns indicate the strength of association of the most significant variant implicating the corresponding gene as causal with respect to the TSH increasing allele, such that shades of blue represent associations with the other thyroid phenotypes that have the same direction of effect as the TSH association and shades of red represent an opposite direction of effect to the TSH association (Supplementary Data 5). As there were no significant pQTL associations, that column has been omitted from the figure.

hyperthyroidism but not hypothyroidism (implicating *TSHR*, *NRG1*, **SDCCAG8**, **HMGA2**, *APOH*, *BMP2*, *SMAD6*), and sentinel variants associated with both hypothyroidism and hyperthyroidism with an opposite direction of effect (implicating *TPO*, **PTEN**, *CAPZB*, *VEGFA*, *NFIA*, *MBIP*, **HLA-C**, *SLC25A37*, *SLC25A37*, **NKX2-1**, *NKX2-3*, *FOXA2*). However, tolerated *SH2B3* missense variant, rs3184504 (allele T), associated with increased TSH, was associated with increased risk of both hypothyroidism and hyperthyroidism, and in our PheWAS with increased risk of other autoimmune disorders and pleiotropic associations with many traits (Supplementary Data 5 and 6). Furthermore, the TSH-increasing allele of *SGK1* intronic SNP rs1743963 was associated with decreased risk of both hypothyroidism and hyperthyroidism and in our PheWAS,

with increased calcium levels. *SGK1*, encoding serum glucocorticoid regulated kinase 1, is involved in the regulation of ion channels and stress response.

Among relevant clinical phenotypes in the PheWAS is "secondary hypothyroidism", defined by PheCode 244.1 which encompasses ICD codes for hypothyroidism due to surgery, ablation or medicaments used in treating hyperthyroidism. This code therefore represents consequences of treated hyperthyroidism—or occasionally treatment of other conditions—and is unrelated to central hypothyroidism, which was sometimes previously referred to as secondary hypothyroidism. We excluded these codes from our definition of hypothyroidism (Fig. 2). When we explored single SNP associations with treated hyperthyroidism we found consistent directions of association with hyperthyroidism clinical codes and instances of associations, implicating genes *IGFBP5*, **CADM1**, *SOX9*, *BMP2*, *TGFB2* and **SYT13**, which were not detected by only studying hyperthyroidism codes in UK Biobank, consistent with earlier onset hyperthyroidism cases (Supplementary Fig. 3; Supplementary Data 7).

Of the TSH sentinel variants implicating putative causal genes, a minority were associated with free T$_4$ (*TPO*, *IGFBP5*, *INSR*, *GLIS3*, *NRG1*, *PDE10A*, *PDE8B*, *CAPZB*, *VEGFA*, *NFIA*, *MBIP*, **HLA-C**, *NEK6*, **CERS6**, **CCBE1**, **GNG7**, **PTPRJ**, **KANK1**, **C9orf156**, **NKX2-1**, *NKX2-3*, *GATA3*), and sentinel variants that were not associated with T$_4$ included sentinel variants associated with hypothyroidism or hyperthyroidism (*TG*, *AKT1*, *TSHR*, *SPATA13*, **CADM1**, *BCAS3*, *SASH1*, **PTEN**, **PDE4D**, *VEGFC*, **SGK1**, **SPPL3**, **SDCCAG8**, **HMGA2**, *NR3C2*, *VAV3*, **CGA**, *SH2B3*, **GNG7**, *APOH*, **TPPP**, *WNT4*, **SNX8**, *BMP2*, **C1orf116**, **KCTD5**, **KDR**, *FOXA2*, **WWTR1**, *TGFB2*, *MAL2*, **ZBTB17**, **ARNT**, *CDC16*). Our findings suggest that the study of TSH levels is a more sensitive approach to detecting genetic associations relevant to thyroid disease than the study of T$_4$ levels.

Genes implicated by a single putative causal missense variant that was deleterious included **SPATA6**, **ADCY6** and *APOH*. *SPATA6* (implicated by rs77303590, minor allele frequency [MAF] 2.6% in EUR) encodes a spermatogenesis-associated protein possibly involved in microfilament transport, and had no associations at FDR <1% in our PheWAS. *ADCY6* (implicated by rs115315671, MAF 2.3% in EUR) encodes an adenylate cyclase protein involved in cAMP signalling, and was associated with creatinine levels in our PheWAS. Apolipoprotein H is involved in lipoprotein metabolism, coagulation and haemostasis. The G allele of the *APOH* missense deleterious variant, rs1801690 (MAF 5.7% in EUR), was associated with reduced TSH, increased risk of hyperthyroidism, increased risk of congenital anomalies of endocrine glands and thyrotoxicosis in the UK Biobank DeepPheWAS analysis, as well as increased aspartate aminotransferase (AST) and alanine aminotransferase (ALT) levels, increased height, reduced triglycerides, reduced carotid intima media thickness and, in FinnGen[16], reduced deep venous thrombosis risk. **C9orf156** (encoding TRNA Methyltransferase O) was implicated by a single putative causal tolerated missense variant, rs2282192 (T allele, frequency 28.8% in EUR), associated with increased TSH, increased hypothyroidism risk and decreased risk of nontoxic multinodular goitre and thyroid cancer as well as lower mean corpuscular volume and HbA1c.

Novel TSH sentinel variants associated with thyroid diseases also implicated relatively understudied putative causal genes, such as **SPPL3** and **SDCCAG8**. *SPPL3* encodes Signal Peptide Peptidase Like 3 involved in T cell receptor signaling, regulation of calcineurin-NFAT signaling and protein dephosphorylation. The **SPPL3** intronic variant rs2393717 G allele (frequency 47.3% in EUR) associated with increased TSH was associated with reduced hypothyroidism risk, increased tyrosine (a thyroid hormone precursor), as well as decreased C-reactive protein, increased insulin-like growth factor 1 (IGF-1), reduced height, whole body fat-free mass and reduced sex hormone binding globulin (especially in males), decreased gamma glutamyltransferase (GGT), increased alkaline phosphatase, reduced platelet count and

eosinophils, increased cholesterol and with lipid composition traits. Mutations in **SDCCAG8**, encoding the sonic hedgehog (SHH) signaling and ciliogenesis regulator, SDCCAG8, cause Bardet-Biedl Syndrome 16 (BBS16). Hypothyroidism and hyperthyroidism have been observed in commoner forms of Bardet-Biedl Syndrome[17]. The TSH increasing allele, C (frequency 53.8% in EUR), of **SDCCAG8** intronic variant rs10926981 was associated with reduced risk of thyrotoxicosis and thyroid cancer, as well as reduced creatinine and increased eGFR.

As smoking is known to influence thyroid function, we conducted a look-up of our sentinel variants in the summary statistics from GSCAN (GWAS & Sequencing Consortium of Alcohol and Nicotine use)[18] across four smoking behaviour traits: age at smoking initiation, cigarettes per day, smoking cessation and smoking initiation. One sentinel variant, rs3184504, implicating a large number of genes including *SH2B3*, was significantly associated with smoking initiation ($P < 5 \times 10^{-8}$); the T allele associated with increased TSH levels was associated with lower risk of smoking initiation[18] (Supplementary Data 5).

## Druggable targets

For the 112 genes supported by ≥2 criteria, we surveyed gene-drug interactions using the Drug Gene Interaction Database (DGIDB). The protein products of these genes include targets for treatments to stimulate (thyrotropin [TSH]) or suppress (methimazole, targeting thyroid peroxidase, TPO) thyroid function, and drugs to treat thyroid cancer (e.g. the KDR inhibitor, vandetanib) as well as PDE4 inhibitors and AKT inhibitors utilised in immunoinflammatory conditions and cancers (Supplementary Data 8).

## Pathway analysis

Employing ConsensusPathDB[19], we tested biological pathways enrichment for the 112 putative causal genes supported by ≥2 criteria, highlighting signal transduction, particularly G protein (Reactome) and cAMP (KEGG) signaling, and the overlapping phosphodiesterases in neuronal function pathway (Wikipathways, including novel genes **PDE4D**, **PDE7A**, **PDE4B**, **ADCY6**). The thyroxine production (Wikipathways) pathway included novel gene **CGA**, encoding anterior pituitary glycoprotein hormones subunit alpha, which is common to TSH, chorionic gonadotropin (CG), luteinizing hormone (LH), and follicle-stimulating hormone (FSH). New pathways of interest include VEGF hypoxia and angiogenesis (Biocarta), including **ARNT**, **BDKRB2**, and **KDR** alongside *VEGFA* and *AKT1*, as well as opioid signalling, and platelet activation (Supplementary Data 9).

## Phenome-wide associations of pathway-specific TSH genetic risk scores

We hypothesised that partitioning a TSH GRS into pathway-specific GRSs according to the biological pathway(s) that each variant influences could inform understanding of mechanisms underlying TSH and thyroid disease, and possible consequences of pathway perturbation. Informed by the above prioritisation of putative causal genes and classification of these genes by pathway, we undertook PheWAS for TSH-weighted GRSs partitioned by each of 26 enriched pathways (FDR $< 5 \times 10^{-4}$) after dropping redundant pathways (GRS correlation $r^2 < 0.7$).

We highlight examples of pathway-specific GRSs showing differing patterns of associations with thyroid and non-thyroid diseases (Supplementary Data 10; Supplementary Fig. 4). The GRS for higher TSH specific to the cAMP signalling pathway (KEGG, including novel genes **GNAS**, **PDE4D**, **PDE4B**, **ADCY6**, **CGA**) was specific to increased risk of hypothyroidism; no associations (FDR <1%) with other traits were shown (Supplementary Fig. 4a). A GRS for higher TSH specific to the activin receptor-like kinase (ALK) in cardiac myocytes pathway (Biocarta, including novel genes **SMAD6**, **BMP4**) showed associations with reduced risk of nontoxic nodular and multinodular goitre, and simple goitre, as well as raised heel bone mineral density, standing height and whole-body fat-free mass and reduced FEV$_1$/FVC (Fig. 3a,

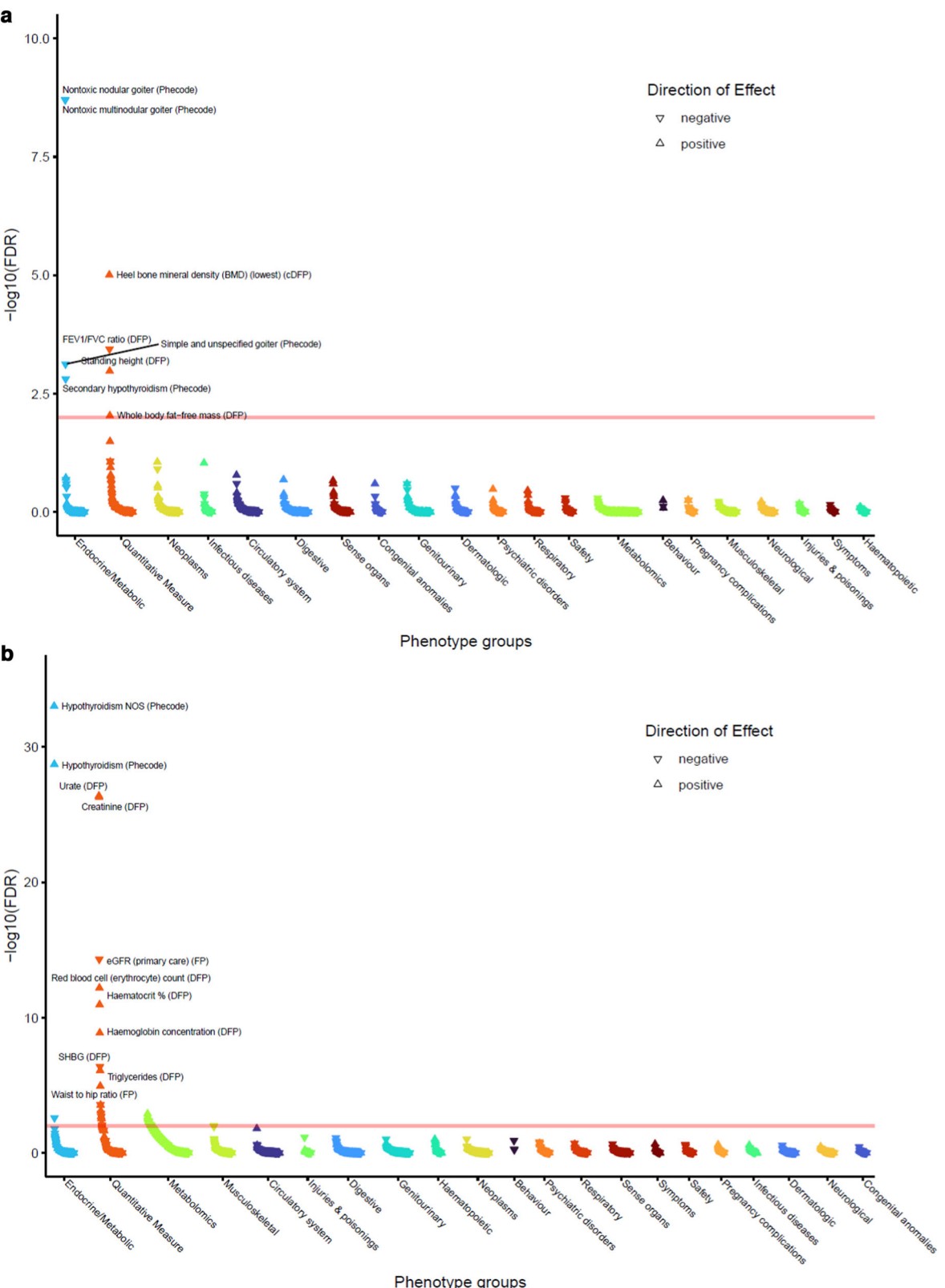

**Fig. 3 | PheWAS for select pathway-specific TSH-weighted GRS.** PheWAS for pathway-specific TSH-weighted GRS partitioned by: (**a**, top) activin receptor-like kinase (ALK) in cardiac myocytes pathway (Biocarta); (**b**, bottom) platelet activation, signalling and aggregation pathway (Reactome).

Supplementary Data 10). GRSs specific to several pathways showed association to PheCode 244.1 capturing consequences of treated hyperthyroidism: the ALK in cardiac myocytes pathway, pathways in cancer (KEGG), factors and pathways affecting IGF-1-Akt signaling (Wikipathways), myometrial relaxation and contraction pathways (Wikipathways), FGFR3 signaling in chondrocyte proliferation and terminal differentiation (Wikipathways). The GRS for higher TSH specific to the platelet activation, signalling and aggregation pathway

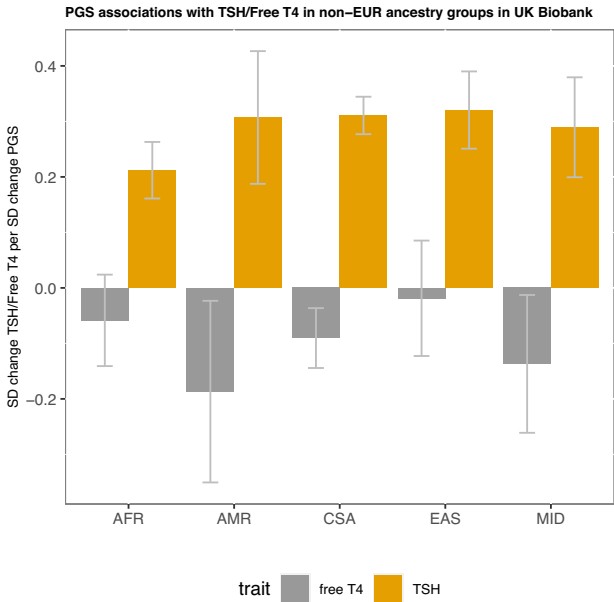

**PGS associations with TSH/Free T4 in non–EUR ancestry groups in UK Biobank**

**Fig. 4 | PGS associations with TSH/free T4 across non-European ancestries in UK Biobank.** The TSH PGS association with TSH and free T4 across ancestry groups in UK Biobank shown as standard deviation (SD) change in TSH/free T4 per SD increase in the PGS. The ancestry groups were as defined by the Pan-UK Biobank initiative – AFR African ancestry, AMR admixed American ancestry, CSA Central/South Asian ancestry, EAS East Asian ancestry, MID Middle Eastern ancestry. Error bars indicate 95% confidence intervals. Sample size for TSH: AFR, $n = 1430$; AMR, $n = 249$; CSA, $n = 3033$; EAS, $n = 721$; MID, $n = 479$. Sample size for free T4: AFR, $n = 584$; AMR, $n = 145$; CSA, $n = 1307$; EAS, $n = 362$; MID, $n = 256$.

(Reactome, including novel genes *GNG7*, *ANXA5*, *PRKCZ*) was associated with increased hypothyroidism risk, reduced nontoxic nodular goitre and simple goitre risk, raised urate, creatinine and reduced eGFR, reduced sex hormone binding globulin and testosterone, increased waist-hip-ratio, handgrip strength and whole body fat-free mass, osteochondropathies, increased triglyceride levels, and with lipid composition traits (Fig. 3b).

### Polygenic score associations

We constructed a PGS for TSH using the summary statistics of approximately 1.12 million SNPs from our Stage 1 analysis (meta-analysed from UK Biobank, EXCEED, and results from Zhou et al. with total sample size of 247,107 European-ancestry individuals) as the training dataset (Online Methods, Fig. 4).

The TSH PGS showed distinct patterns of associations with relevant thyroid and non-thyroid phenotypes in our PheWAS (Supplementary Fig. 5, Supplementary Data 11). Thyroid-relevant PGS associations included increased risk of hypothyroidism, lower risk of non-toxic (multi)nodular goitre, thyrotoxicosis, Graves' disease, and thyroid cancer, and reduced tyrosine. Other PGS associations included increased FEV$_1$/FVC, lower risk of chronic obstructive pulmonary disease (COPD), pneumonia, coeliac disease, common cancers and multisite chronic pain, lower arterial stiffness, increased creatinine and urate, increased alkaline phosphatase and aspartate aminotransferase, increased eosinophils, decreased sex hormone-binding globulin, testosterone and IGF-1, decreased glucose (Supplementary Data 11) as well as altered lipid levels and composition. We found little or no attenuation of these PGS associations after adjustment for whether the individuals had ever smoked (Supplementary Data 12).

We then tested PGS associations across ancestries in UK Biobank. Strong associations were shown with TSH levels in all ancestry groups tested (African, AFR; Admixed American, AMR; Central/South Asian, CSA; East Asian, EAS; Middle Eastern, MID; Fig. 4, Supplementary

Data 13). The TSH PGS was strongly associated with free T$_4$ levels in European ancestry individuals (SD change in phenotype per SD change in PGS ($\beta$) = −0.0704; 95% confidence interval (CI): [−0.0790, −0.0618]; $P = 5.93 \times 10^{-58}$), nominally associated with free T$_4$ in the next largest ancestry group, CSA ($\beta = -0.0904$; 95% CI: [−0.1443, −0.0365]; $P = 0.0010$, 1307 participants), and showed a consistent direction of effect in other ancestry groups (Supplementary Data 13).

To inform understanding of the relevance of the TSH PGS for disease, we subsequently tested disease susceptibility risk in all UK Biobank ancestry subgroups with at least 100 cases. In European ancestry UK Biobank participants, the TSH PGS was associated with risk of hypothyroidism (odds ratio per SD change in PGS (OR) = 1.46; 95% CI: [1.44,1.48]; $P < 1 \times 10^{-300}$), hyperthyroidism (OR = 0.67; 95% CI: [0.65,0.69]; $P = 9.17 \times 10^{-166}$), thyroid cancer (OR = 0.78, 95% CI: [0.72, 0.85]; $P = 7.20 \times 10^{-10}$) and other thyroid disease (OR = 0.69; 95% CI: [0.65,0.72]; $P = 1.35 \times 10^{-39}$, Supplementary Data 14). In other ancestry groups, a consistent direction of association was shown with each of these traits, the largest case numbers being seen for hypothyroidism in CSA (862 cases; OR = 1.42; 95% CI: [1.32,1.53]; $P = 9.45 \times 10^{-21}$, Supplementary Data 14). Results from a sensitivity analysis in an independent subset (i.e. excluding individuals who were included in our Stage 1 analysis or share at least 2nd degree relatedness with individuals included in the Stage 1 analysis) were consistent with these findings (Supplementary Data 15). We then tested the PGS in independent South Asian population from Genes & Health, in which the TSH PGS was associated with hypothyroidism (OR = 1.41; 95% CI: [1.35,1.46]; $P = 1.51 \times 10^{-68}$) and other thyroid disease (OR = 0.83; 95% CI: [0.78,0.89]; $P = 4.92 \times 10^{-8}$) (Supplementary Data 14).

To further understand the clinical relevance of the PGS we examined risk of thyroid disease per decile of the PGS in European ancestry individuals. Individuals in the highest decile had 3.65-fold higher odds (95% CI: [3.42,3.90]; $P < 1 \times 10^{-300}$) of hypothyroidism compared with those in the lowest decile, whilst those in the lowest decile had 4.21-fold (95% CI: [3.67,4.83]; $P = 3.78 \times 10^{-92}$) and 2.18-fold higher odds (95% CI: [1.52,3.14]; $P = 2.77 \times 10^{-5}$) of hyperthyroidism and thyroid cancer, respectively, and 3.40-fold higher odds (95% CI: [2.59,4.46]; $P = 1.11 \times 10^{-18}$) of other thyroid disease compared with those in the highest decile (Fig. 5, Supplementary Data 14). The prediction performance was evaluated using receiver operating characteristic (ROC) curves, with the area under the curve (AUC) for hypothyroidism, hyperthyroidism, thyroid cancer and other thyroid disease being 71.6% [71.3%–71.9%], 73.5% [72.7%–74.3%], 64.6% [62.5%–66.7%] and 76.0% [74.6%–77.3%], respectively, when age and sex were combined with TSH PGS (Supplementary Fig. 6).

Given questions about how best to deploy and repeat testing for thyroid disease in asymptomatic patients and in patients with non-specific symptoms, we explored whether membership of a high or low risk decile for TSH PGS was associated with differences in age of onset of hypothyroidism or hyperthyroidism (Online Methods). Between individuals with median, highest and lowest deciles of the TSH PGS, clear differences were seen in age of onset of hypothyroidism ($P < 1.0 \times 10^{-300}$, Fig. 6a) and hyperthyroidism ($P = 4.30 \times 10^{-61}$, Fig. 6b). For example, a 5% prevalence of hypothyroidism was reached by age 51.1 years in the highest TSH PGS decile versus by 74.7 years in the lowest TSH PGS decile. Similarly a 1% prevalence of hyperthyroidism was reached by age 47.3 years in the lowest PGS decile compared with 71.2 years in the highest TSH PGS decile. Results from sensitivity analyses excluding individuals who were included in our Stage 1 analysis or share at least 2nd degree relatedness with individuals included in the Stage 1 analysis were consistent with the findings described here (Supplementary Figs. 7, 8 and 9).

### Discussion

The large sample size of our study, achieved through utilising quality-controlled TSH measures from UK primary health care

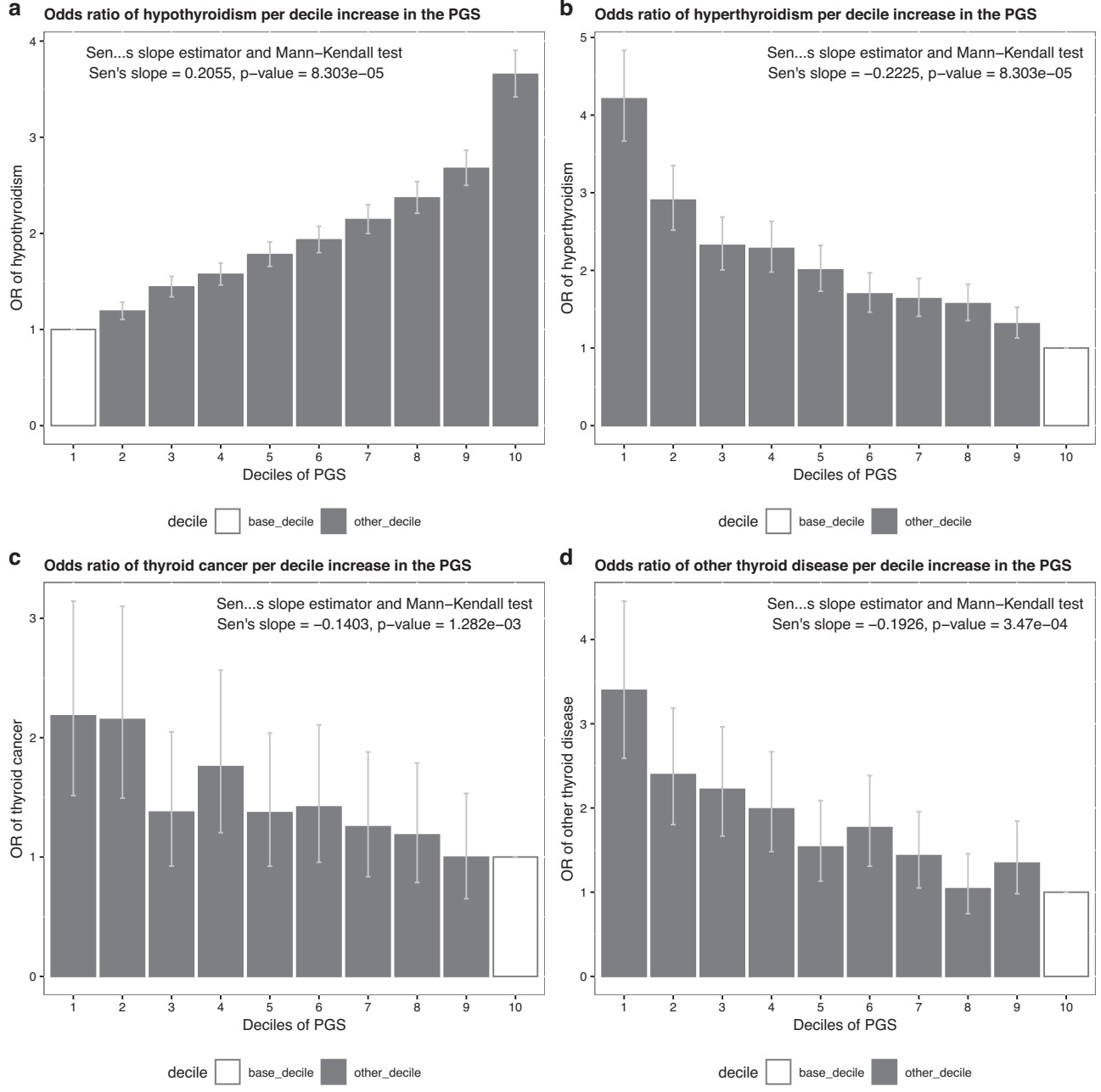

**Fig. 5 | Association of TSH PGS deciles with thyroid diseases.** The TSH PGS decile analysis for four clinical thyroid phenotypes—(**a**, top left) hypothyroidism, (**b**, top right) hyperthyroidism, (**c**, bottom left) thyroid cancer, and (**d**, bottom right) other thyroid disease. Statistical tests were two-sided, the height of the bars show the point estimate of the effect and whiskers show the 95% CI. OR odds ratio. The Mann- Kendall test is a test for monotonic trend. Sample size: Hypothyroidism, 29,550 cases and 368,691 controls; Hyperthyroidism, 5549 cases and 392,692 controls; Thyroid cancer, 624 cases and 358,144 controls; Other thyroid disease, 1233 cases and 124,515 controls.

records, increased the yield of TSH sentinel variants by over 2.5-fold, to 260. Through the most comprehensive initiative to identify putative causal variants and genes for TSH levels, we defined 112 genes implicated by multiple criteria. This is the first study to develop pathway-specific GRS for TSH levels and to use these in PheWAS, through our new DeepPheWAS platform[7], to investigate the potential consequences of intervening in relevant pathways. It is also the first study to visualise and compare the incidence of hypothyroidism and hyperthyroidism over time among individuals grouped according to their TSH PGS, showing marked differences in ages of onset of these conditions according to PGS deciles.

We implicate novel putative causal variants and genes, which alongside those previously reported[6,15,20], provide a more complete picture of relevant pathways and putative mechanisms. Pathways we highlight include signal transduction and cAMP signaling, as well as pathways not confidently implicated previously such as VEGF hypoxia and angiogenesis, AKT signaling, and platelet activation. Our findings are consistent with signaling or response to thyroid or non-thyroid hormones (including IGF-1 signaling), neuronal protection, angiogenesis and ciliogenesis influencing TSH levels and thyroid diseases.

Pleiotropic effects of aggregated TSH-associated variants have been previously shown through PheWAS. Partitioning TSH-associated

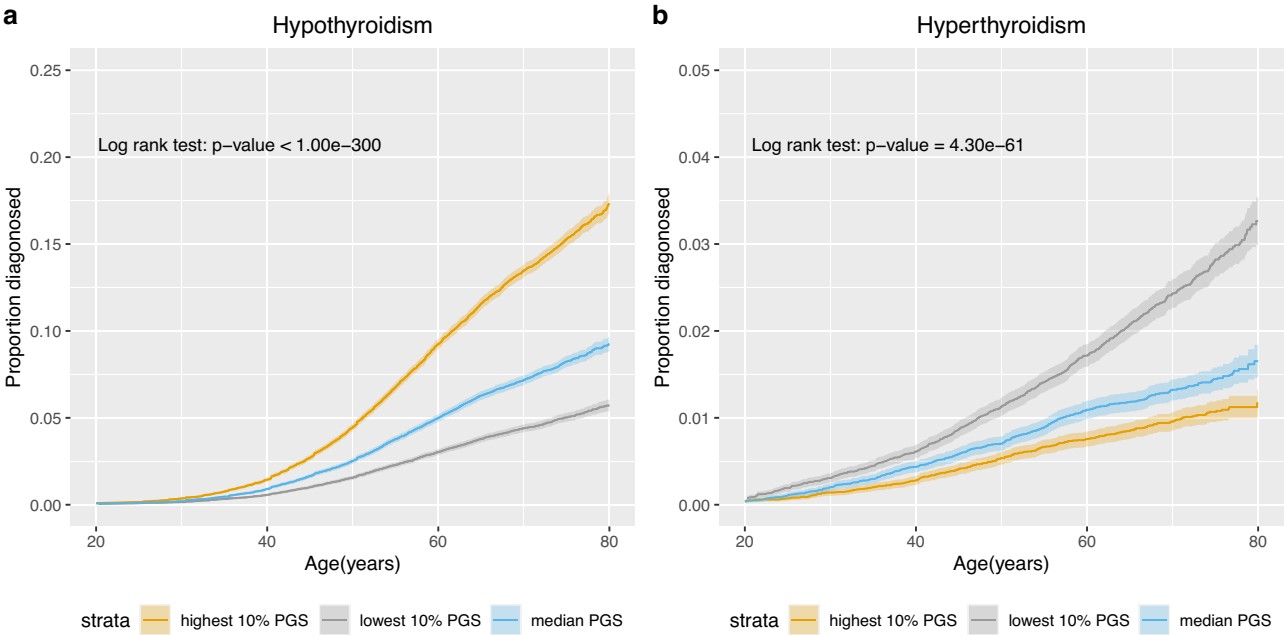

**Fig. 6 | Association of TSH PGS with age-of-onset of hypothyroidism and hyperthyroidism.** Proportion of hypothyroidism (**a**, left) and hyperthyroidism (**b**, right) cases diagnosed by age stratified into lowest (grey), median (blue) and highest (yellow) decile for the TSH PGS. Shaded bands indicate 95% confidence intervals. Sample size: Hypothyroidism, 29,550 cases and 368,691 controls; Hyperthyroidism, 5549 cases and 392,692 controls. Statistical tests were two-sided.

variants by pathway provides a more nuanced understanding of the consequences of pathway perturbation on thyroid and non-thyroid disorders. We show contrasting patterns of phenotype association—for example highly specific associations for hypothyroidism (cAMP signalling) versus associations also with body composition, renal function and lipid traits (platelet activation pathways). As individuals may have high GRS for one or more pathways and low GRS for other pathways[21], individuals' pathway GRS profiles may relate to patterns of comorbidities, and could have implications for treatment choices in thyroid diseases.

Here we adopted a powerful strategy for discovery of genetic variants associated with thyroid diseases. We studied TSH as a quantitative measure within reference ranges, and detected novel sentinel variants which individually and in aggregate are associated with thyroid diseases. Not all TSH-associated variants showed association with free T₄ levels, even those associated with thyroid disease, highlighting the value of TSH as a sensitive marker of thyroid disorders. In addition, thyroid function within the euthyroid range is associated with adverse outcomes[22] and thus TSH-associated variants that have not yet been overtly associated with thyroid disease remain highly relevant. GWAS of other quantitative traits—including those on the UK Biobank biomarker panel—have highlighted a number of targets leading to active drug development for related diseases[3–5]. However, TSH has not yet been measured in UK Biobank samples. Thus, harnessing TSH levels measured in primary care in the EXCEED and UK Biobank studies more than doubled available sample sizes.

The PGS we developed for TSH measures genome-wide genetic risk, using association statistics from the Stage 1 analysis. To our knowledge this is the first TSH PGS to be shown to be associated with TSH levels across all ethnic groups in UK Biobank. Power was more limited for testing other traits and diseases in the much smaller sample sizes available in non-European ancestries in UK Biobank, but there was a strong association with hypothyroidism ($P = 1.51 \times 10^{-68}$) and other thyroid disease ($P = 4.92 \times 10^{-8}$) in South Asian participants from the independent Genes & Health study. Understanding the genetic architecture of thyroid diseases within and across ancestries requires larger sample sizes in non-European ancestries, and an urgent global effort is

required to include much more diverse populations in genomic studies than has been the case to date[23].

The PGS shows a strong association with age of onset of hypothyroidism and hyperthyroidism in European ancestry individuals. Universal screening for thyroid disease is not recommended[24,25]. Instead, case finding strategies are adapted to personal risk factors such as age, family history and relevant long-term conditions. Our findings raise the possibility of tailoring case finding strategies for thyroid disease according to a PGS for thyroid disease, especially if genome-wide data become available as part of the medical record. Further development and testing of a PGS would be required in independent, diverse populations, and alongside risk factors already employed in case finding.

Our study has potential limitations. Stage 1 focused on European ancestry individuals and the overall design included too few studies and participants of non-European ancestry to quantify heterogeneity in allelic effects on TSH attributable to ancestry using transethnic meta-analyses[26]. Non-European ancestry populations are markedly under-represented in genomic studies globally, requiring diversification of new studies[23]. As with other contemporary genome-wide association meta-analysis, maximising power for discovery leaves fewer independent datasets for follow-up. However, we were able to include 96,497 additional participants (including 33,171 individuals of South Asian ancestry) in Stage 2 and show that after meta-analysis of Stages 1 and 2 that 230 of 249 sentinels available in Stage 2 still met genome-wide significance. Whilst we replicated associations in UK Biobank for 91 of 98 available variants from the largest previous meta-analysis of TSH, we cannot rule out selection bias. For example, effect estimates could be biased if research participants in whom TSH levels were routinely measured differed from participants in which they were not measured, although 78% of UK Biobank participants with linked primary care data had one or more TSH measures. Our approach of excluding measurements of TSH outside the normal range could miss some variants more relevant to the extremes of the distribution. Effect estimates could be attenuated through previous treatment for thyroid diseases. As thyroid treatment is unlikely to commence before a TSH measure, we used only the first available TSH measure. Effect size

estimates for disease associations of TSH-associated variants could also be attenuated due to misclassification of thyroid diseases in EHRs. To mitigate this, we undertook careful curation of clinical codes used to define clinical thyroid disease, such as exclusion of cases of hypothyroidism resulting from treatment of hyperthyroidism. To avoid classifying cases incorrectly as hyperthyroidism or hypothyroidism we defined a separate category "other thyroid disease" containing disorders for which thyroid activity levels were less clear, such as goitre and thyroiditis. In the latter group, the PGS associations were in the same direction as for hyperthyroidism, with a smaller effect estimate.

A limitation of our variant-to-gene mapping approach is the equal weighting given to all lines of variant-to-gene evidence, however strategies for in silico mapping of associated variants to causal genes are evolving and difficult to evaluate without a reference set of fully functionally characterized variants and causal genes.

In summary, we more than doubled the number of TSH-associated sentinel variants to 260, implicated 112 priority genes, showed their relevance to thyroid diseases, and developed pathway-specific genetic risk scores which show differential patterns of pleiotropy of relevance in understanding co-morbidities and treatment choices. We also demonstrate the relevance of our results to a range of ancestries and highlight the need for better representation of all ancestries in the future study of thyroid genetics. The PGS we developed was associated with risk of thyroid disease and with age of onset of hypothyroidism and hyperthyroidism, and suggests potential utility of genetic information in future case-finding strategies, subject to further development and appropriate evaluation.

## Methods

### Ethical approval
The UK Biobank genetic and phenotypic data were analysed under UK Biobank Application 43027. UK Biobank has ethical approval from the UK National Health Service (NHS) National Research Ethics Service (11/NW/0382). EXCEED received ethical approval from the Leicester Central Research Ethics Committee (13/EM/0226). Genes & Health received ethical approval from the NRES Committee London – South East (14/LO/1240).

The activities of the EstBB are regulated by the Human Genes Research Act, which was adopted in 2000 specifically for the operations of EstBB. Individual level data analysis in EstBB was carried out under ethical approval 1.1-12/624 from the Estonian Committee on Bioethics and Human Research (Estonian Ministry of Social Affairs), using data according to release application 6-7/GI/2013 from the Estonian Biobank.

Informed consent was obtained from all participants.

### Cohort details
UK Biobank is a cohort of approximately 500,000 individuals recruited from across the United Kingdom[27]. Individuals aged between 40 and 69 years were recruited from the general population between 2006 and 2010.

The Extended Cohort for E-health, Environment and DNA (EXCEED) recruited ~10,000 individuals primarily through local general practices in Leicester City, Leicestershire and Rutland[28]. Recruitment starting in 2013. Individuals invited to contribute to EXCEED were aged between 40 and 69 years.

Estonian Biobank is a population-based biobank with 212,955 participants in the current data freeze (2023v1). All biobank participants have signed a broad informed consent form and information on ICD codes is obtained via regular linking with the national Health Insurance Fund and other relevant databases, with a majority of the electronic health records having been collected since 2004[9].

Genes & Health is a cohort of ~50,000 British Pakistani and Bangladeshi individuals recruited primarily in East London, England[10]. Individuals aged 16 years and over were recruited via community settings, such as places of worship, markets, and libraries, or from healthcare settings, such as primary care practices and outpatient clinics.

The Zhou et al[6]. study is a meta-analysis of three constituent studies: (i) the HUNT study, a population-based study of around 120,000 individuals aged 20 years and over conducted in the county of Nord-Trøndelag, Norway, since 1984[29]; (ii) the Michigan Genomics Initiative (MGI), which recruited participants awaiting diagnostic or interventional procedures in the Michigan Medicine health system, and (iii) the ThyroidOmics consortium, a collection of 22 independent US- and Europe-based cohorts in approximately 55,000 individuals.

All studies contributing to our analysis collected information at baseline concerning lifestyle and health outcomes, as well as providing linkage to electronic health records and genetic data. Further details of genotyping and analysis are described in the Supplementary Note.

### Phenotype
We captured all TSH results reported in the primary care data available in up to 230,000 individuals in UK Biobank and 8500 individuals in EXCEED utilising codes from Read version 2 and Read version 3 (Clinical Terms Version 3 or "CTV3", Supplementary Data 16). We took an individual's first non-missing TSH measurement to minimise the effect of thyroid function-altering medications on our phenotype as an individual is unlikely to have received these medications before their first thyroid function test. We excluded individuals with a TSH measurement <0.4 or >4.0 mU/L as has been done previously[15]. TSH values were captured following the same strategy in the Estonian Biobank and Genes & Health cohort. In Zhou et al., TSH values were captured using varying approaches across the constituent cohorts (described further in the Supplementary Note) and we used publicly available summary statistics for our analysis.

For the 260 sentinel variants, we compared the −log10P values from association analyses using the whole range of TSH measures (unrestricted) versus using the restricted range and we found generally less significant P-values for the unrestricted range (Supplementary Fig. 10, Supplementary Data 2), suggesting we did not lose statistical power with our approach.

### Stage 1 analysis
In UK Biobank and EXCEED, we applied an inverse normal transformation to the residuals from linear regression of the TSH phenotype against age (at time of measurement) and sex. This transformed phenotype was used for genome-wide association testing under an additive genetic model adjusted for age, genotyping array, sex and the first 10 principal components of ancestry using PLINK 2.0[30], generating effect size estimates (β), standard errors, and P-values. We analysed individuals of European ancestry, as defined by the Pan-UK Biobank initiative[31], who were not more closely related than third-degree relatives using a KING software relatedness coefficient >0.0884 to indicate second-degree relatives or closer[32]. UK Biobank was imputed to the Haplotype Reference Consortium and merged UK10K and 1000 Genomes phase 3 reference panels[11] and we tested genetic variants with a minor allele count >20 and imputation score >0.5. EXCEED was imputed to the TOPMed (https://topmed.nhlbi.nih.gov/) panel and due to the smaller sample size, we tested genetic variants with a minor allele frequency >0.1% and imputation score >0.5.

We derived the LD Score regression intercept using LDSC v1.0.1[33] to estimate inflation in our test statistics due to confounding, such as by cryptic relatedness or population stratification. We estimated, separately, the LD Score regression intercept for the GWAS in UK Biobank and EXCEED. The UK Biobank test statistics were corrected for inflation ($\lambda_{LDSC} = 1.05$) prior to meta-analysis. The EXCEED test statistics were not corrected for inflation ($\lambda_{LDSC} = 0.98$).

We used METAL v2018-08-28[34] to conduct a fixed-effect inverse variance-weighted meta-analysis of the GWAS in UK Biobank and

EXCEED and the previous largest GWAS of TSH[6]. We describe the meta-analysis of UK Biobank, EXCEED and the previous largest GWAS as Stage 1. Since the EXCEED results were aligned to GRCh38, we ran LiftOver v2011-09-27[35] to map the results to GRCh37. Over 99.5% of the genetic variants tested in the GWAS in EXCEED were successfully mapped to GRCh37. Following meta-analysis, we estimated the LD Score regression intercept once more ($\lambda_{\mathrm{LDSC}} = 1.00$).

We estimated the proportion of variance explained by the sentinel SNPs using the formula:

$$\frac{\sum_{i=1}^{n} 2f_i(1-f_i)\beta_i^2}{V} \tag{1}$$

where $n$ is the number of SNPs, $f_i$ and $\beta_i$ are the frequency and effect estimate of the $i^{\mathrm{th}}$ variant from our Stage 1 analysis, and $V$ is the phenotypic variance.

## Sentinel variant selection and fine mapping

Using Stage 1 results only, we selected 2 Mb loci centered on the most significant variant for all regions containing a variant with $P < 5 \times 10^{-8}$. Loci within 500 kb of each other were merged for fine mapping. Annotation-informed fine mapping in each locus defined statistically independent credible sets (see below) and sentinel variants were defined as the variant in each credible set with the highest posterior probability.

PolyFun v2022-01-27[36] and Sum of Single Effects (susieR) v0.12.27[37] was used to fine-map autosomal non-HLA loci utilising pre-computed functional prior causal probabilities based on a meta-analysis of 15 UK Biobank traits. The functional priors are proportional to per-SNP heritabilities estimated from the functional enrichments of 187 variant annotations from the baseline-LF 2.2.UKB model[38], including those relating to conservation, regulation, MAF and linkage disequilibrium (LD), which were estimated using an extension of stratified-LDSC[39]. Imputed genotype data from 10,000 randomly selected European individuals from UK Biobank was used as an LD reference. Loci for which PolyFun and SuSiE did not identify any credible sets, as well as HLA and chromosome X loci, were fine-mapped using the Wakefield method[40] with the prior $W$ set as 0.04 in the approximate Bayes factor formula. 95% credible sets were generated for all loci, by adding variants in descending order of posterior inclusion probability (PIP) until the sum total PIP in the set reaches 95% such that we have 95% confident the causal variant is contained in the set. Variants with the highest PIP per credible set are listed (Supplementary Data 17).

## Stage 2 analysis

We took sentinel variants reaching $P < 5 \times 10^{-8}$ in Stage 1 into Stage 2, in which TSH associations were tested in 63,326 EUR participants from the Estonian Biobank[9] and 33,171 South Asian ancestry (SA) participants from Genes & Health[10].

Using METAL v2018-08-28, we performed a fixed-effect inverse variance-weighted meta-analysis of the results from the Estonian Biobank and Genes & Health. We subsequently meta-analysed Stages 1 and 2, using METAL (described above). We used $P < 5 \times 10^{-8}$ in the meta-analysis of Stages 1 and 2 to highlight sentinel variants that remained significant after inclusion of independent datasets. All included studies used the same approach to covariate adjustment as applied in Stage 1.

## Novel sentinel variants

We searched PubMed and GWAS Catalog to identify applied studies focused on thyroid stimulating hormone and associations reaching $P < 5 \times 10^{-8}$. These included the following sources: Gudmundsson et al.[41], Kwak et al.[42], Malinowski et al.[43], Medici et al.[44], Nielsen et al.[45], Popović et al.[46], Porcu et al.[20], Taylor et al.[47], Teumer et al.[15], and Zhou

et al.[6]. We determined whether a sentinel variant was novel if its extent of LD with nearby previously reported sentinel variants was $r^2 < 0.2$.

## Associations with clinical thyroid disease

We tested the association between our sentinel variants and free thyroxine ($T_4$). We then tested the association between our sentinel variants and four clinical thyroid diseases: (i) hypothyroidism; (ii) hyperthyroidism; (iii) thyroid cancer; and (iv) other (non-cancer) thyroid disease. Using the UK Biobank primary care data, we extracted free $T_4$ measurements that co-occurred with the corresponding individual's first TSH measurement. To maximise power for hypo- and hyperthyroidism, we identified all potential cases in the primary care data using specific Read codes, in the secondary care data using specific ICD-9/10 codes, and in the self-reported diagnostic data (UK Biobank Data-Field 20002, code 1225 for "hyperthyroidism/thyrotoxicosis" and code 1226 for "hypothyroidism/myxoedema"). For the remaining clinical disease phenotypes, we defined these using specific Read codes in primary care alone (other thyroid disease), and specific ICD-9/10 codes in cancer register data (thyroid cancer). To reduce the overlap in cases for the clinical disease phenotypes, we defined a case by their first diagnosis of hypothyroidism, hyperthyroidism, thyroid cancer and other thyroid diseases. We defined controls for these conditions as any UK Biobank participant who was not defined as a case. When assessing relatedness between cases and controls, we preferentially excluded controls to maintain the maximum possible number of cases. The clinical codes used to define free $T_4$ and the clinical thyroid diseases are presented in (Supplementary Data 16). We used PLINK 2.0[30] to test the associations using logistic regression under an additive genetic model adjusted for sex, genotyping array, age at recruitment to UK Biobank and the first 10 principal components of ancestry. The free $T_4$ phenotype was inverse-normal transformed in the same manner as the TSH phenotype, and its association with our TSH sentinel variants was assessed using linear regression.

These phenotypes were further tested for association with our polygenic score (described below).

## Identification of putative causal genes

To systematically prioritise putative causal genes for TSH-associated variants, we integrated eight sources of evidence including: (i) the nearest gene to the sentinel variant; (ii) the gene with the highest polygenic priority score (PoPS)[12], a method based on the assumption that causal genes on different chromosomes share similar functional characteristics; identification of (iii) expression quantitative trait loci (eQTLs) or (iv) protein quantitative trait loci (pQTLs) within the credible sets; (v) proximity to a gene for a thyroid-associated Mendelian disease (±500 kb); (vi) an annotation-informed credible set containing a missense/deleterious/damaging variant with a posterior probability of association >50%; (vii) identification of a rare variant (±500 kb of a TSH sentinel variant) association with hypo- or hyperthyroidism using whole-exome[13] and whole-genome[14] sequencing resources; and (viii) proximity to a human ortholog of a mouse knockout gene with a thyroid-related phenotype (±500 kb).

For (i), (v), (vii) and (viii), all 260 sentinel variants were used as input; for (ii), the 257 autosomal sentinel variants were used as input; for (iii), (iv) and (vi), all variants across the credible sets were used as input. Where there were two sentinel variants with the joint-highest posterior probability in their credible set, the one with the smallest P-value was used.

We catalogued previously reported genes (Supplementary Data 4) implicated by mapping genome-wide significant sentinels for thyroid traits using eQTL colocalization ($P < 1 \times 10^{-7}$)[15] or DEPICT (FDR ≤ 0.01, ref. [6]), to define whether the genes we implicated were novel.

**Expression quantitative trait loci (eQTLs).** We used the SNP2GENE function implemented by FUMA v1.5.4[48] to facilitate the eQTL analysis.

FUMA contains several eQTL datasets across a broad range of tissue types. We ran SNP2GENE requesting eQTL results from the GTEx v8 (thyroid, hypothalamus and pituitary tissues) and eQTLGen (blood, cis- and trans-eQTLs) datasets. We performed approximate colocalisation between our GWAS and eQTL sentinel variants by identifying whether a sentinel eQTL SNP was in one of our 95% credible sets (Supplementary Data 18).

**Protein quantitative trait loci (pQTLs).** Two pQTL datasets were included in the pQTL analyses: deCODE Genetics[49], with data for 4719 proteins measured by 4907 aptamers, and the SCALLOP Consortium[50], including 90 cardiovascular proteins. The significance level for pQTL associations were set as in the original publications: $P$-value $< 1.8 \times 10^{-9}$ for deCODE Genetics[49] and $P$-value $< 5 \times 10^{-8}$ for the SCALLOP Consortium[50]. We performed approximate colocalisation between our GWAS and pQTL sentinel variants by identifying whether a sentinel pQTL SNP was in one of our 95% credible sets.

**Polygenic priority score (PoPS).** We used a gene prioritization tool, PoPS[12], to calculate gene features enrichment based polygenic priority score[50] to predict genes for our TSH sentinel variants. The full set of gene features used in the analysis included 57,543 total features – 40,546 derived from gene expression data, 8718 extracted from a protein–protein interaction network, and 8479 based on pathway membership. In this study, we prioritized genes for all autosomal TSH sentinel variants within a 500 kb (±250 kb) window of the sentinel variant and reported the top prioritised genes in the region. If there was no gene prioritized within a 500 kb window of the sentinel, we reported any top prioritized genes within a 1 Mb window (Supplementary Data 19).

**Nearby Mendelian disease genes.** We selected rare Mendelian-disease genes from ORPHANET (https://www.orpha.net/) within ±500 kb of a TSH sentinel variant that were associated with thyroid-related diseases. We implicated the gene if the string "thyro" (but not "parathyro") was included in either the disease name or appeared frequently in human phenotype ontology (HPO) terms for that disease. We manually checked the diseases and HPO terms identified for relevance (Supplementary Data 20).

**Nearby mouse knockout orthologs with thyroid related phenotype.** We selected human orthologs of mouse knockout genes with thyroid-related phenotypes, as listed in the International Mouse Phenotyping consortium (https://www.mousephenotype.org/) within ±500 kb of a TSH sentinel variant. The thyroid-related phenotypes included enlarged thyroid gland, abnormal thyroid gland morphology and increased/decreased circulating thyroxine level (Supplementary Data 21).

**Functional annotation of credible sets.** We annotated variants in the 95% credible sets using Variant Effect Predictor (VEP)[51]. We implicated the gene if there was a variant with >50% posterior probability in the credible set that was also either a missense variant, annotated as "deleterious" by SIFT, annotated as "damaging" by PolyPhen-2 or had a CADD PHRED score ≥20.

**Rare variant analysis from whole-exome and whole-genome sequencing**
We performed a lookup for rare variant associations with hypothyroidism or hyperthyroidism within ±500 kb of a TSH sentinel variant using the following resources: (i) single variant and gene-based exonic associations from the AstraZeneca PheWAS Portal[13] (https://azphewas.com/); (ii) single variant whole-genome associations in 150,119 UK Biobank participants[14]. For all tests, we used MAF < 1% and $P < 5 \times 10^{-6}$ (Supplementary Data 22).

**Pathway analysis**
We used ConsensusPathDB[19] to test for enrichment of our prioritised genes in up to 31 pathway and gene set ontology databases. Pathways with FDR < 5% are reported.

**Pathway-specific GRS**
We selected 26 pathways that were enriched at FDR $< 5 \times 10^{-4}$ for our 112 genes implicated by 2 or more lines of evidence (Supplementary Data 9). We created a weighted GRS (weights estimated from the TSH meta-analysis of UK Biobank and EXCEED) for each of the 26 pathways by including, for each gene in the pathway, the variant with the most significant $P$-value that implicates the gene in our variant-to-gene mapping (Supplementary Data 3). Each of the 26 GRS were then checked for association with up to 1939 traits in the PheWAS.

**Polygenic score (PGS)**
We applied PRS-CS-auto v1.0.0[52] to construct a polygenic score (PGS) using the summary statistics from our Stage 1 analysis as the training dataset. PRS-CS-auto is a Bayesian approach, which automatically learns hyper-parameters from the training data; no validation dataset is required. We tested the association of this PGS trained from EUR ancestry group with TSH in non-EUR ancestry groups in UK Biobank, including AFR, AMR, CSA, EAS, MID. Associations were tested using a linear regression model, adjusted for genotyping array, age at TSH measurement, sex and the first 10 principal components of ancestry. We evaluated the association of the TSH PGS with susceptibility to hypothyroidism, hyperthyroidism, thyroid cancer, other thyroid diseases and thyroid eye disease in ancestry groups with more than 100 cases in UK Biobank, and Genes & Health. Associations were tested using logistic regression models, adjusted for genotyping array, sex and the first ten 10 principal components of ancestry. To further aid clinical interpretation, we divided individuals into deciles according to their PGSs and using logistic regression, investigated disease risk associated, comparing each decile to a reference decile. To evaluate the prediction performance, we used ROC curve and assessed the AUC for thyroid relevant diseases. To evaluate the age-dependent PGS performance, we used the Kaplan–Meier method to generate a cumulative incidence plot and a log-rank test to test for differences between groups. In a sensitivity analysis, we assessed the possible impact of overfitting when testing the association between the PGS and binary disease phenotypes by excluding individuals used in the discovery TSH GWAS in UKB and closely related individuals (KING software relatedness coefficient >0.0884) from the testing datasets.

**Phenome-wide association study (PheWAS)**
To identify pleiotropic associations with a wide range of phenotypes, we used DeepPheWAS v0.2.9[7], a flexible PheWAS framework which incorporates phenotypes not present in other PheWAS platforms for: (i) sentinel variants implicating genes supported by ≥3 variant-to-gene mapping criteria (Supplementary Data 3), variants in a credible set that were annotated as missense/damaging/deleterious/phred-scaled CADD score ≥20 that also had a posterior probability >50% (Supplementary Data 23), and low-frequency sentinel variants (MAF < 1%, Supplementary Data 2); (ii) the PGS for TSH; (iii) pathway-specific genetic risk score (GRS).

**Druggability**
To identify gene products that are the targets of drugs, we queried the Drug Gene Interaction Database (DGIDB) (https://www.dgidb.org) for the 112 putative causal genes supported by ≥2 variant-to-gene criteria (Supplementary Data 3). Genes were mapped to ChEMBL interactions and indications (MeSH headings).

**Reporting summary**

Further information on research design is available in the Nature Portfolio Reporting Summary linked to this article.

## Data availability

Access to UK Biobank (https://www.ukbiobank.ac.uk/), Estonian Biobank (https://genomics.ut.ee/en/content/estonian-biobank), Genes & Health (https://www.genesandhealth.org/) and EXCEED (https://exceed.org.uk/) datasets is available to bona fide researchers upon application (in accordance with the terms of ethical approval and participant consent). Further information on how to access the data is available from the study websites. The genome-wide summary statistics from Zhou et al. (2020) can be freely downloaded from http://csg.sph.umich.edu/willer/public/TSH2020/. Genome-wide summary statistics will be made publicly available via the EMBL-EBI GWAS Catalog. URLs for other external datasets are as follows: GTEx (https://www.gtexportal.org/home/), eQTLGen (https://www.eqtlgen.org/), FUMA (https://fuma.ctglab.nl/), deCODE (https://www.decode.com/), SCALLOP (https://olink.com/our-community/scallop/), ConsensusPathDB (http://cpdb.molgen.mpg.de/), Ensembl Variant Effect Predictor (https://www.ensembl.org/info/docs/tools/vep/index.html), the Drug Gene Interaction Database (https://www.dgidb.org/), PoPS (https://github.com/FinucaneLab/pops), Orphanet (https://www.orpha.net/), the International Mouse Phenotyping Consortium (https://www.mousephenotype.org/), PolyFun (https://github.com/omerwe/polyfun), PubMed (https://pubmed.ncbi.nlm.nih.gov/), EMBL-EBI GWAS Catalog (https://www.ebi.ac.uk/gwas/), DEPICT (https://github.com/perslab/depict), AZ PheWAS portal (https://azphewas.com/), genome assembly GRCh37 (https://www.ncbi.nlm.nih.gov/datasets/genome/GCF_000001405.13/).

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

## Acknowledgements

The research was partially supported by the NIHR Leicester Biomedical Research Centre and through an NIHR Senior Investigator Award to M.D.T.; views expressed are those of the author(s) and not necessarily those of the NHS, the NIHR or the Department of Health. The funders had no role in the design of the study. This research was funded in whole, or in part, by the Wellcome Trust: Wellcome Trust Investigator Award (WT202849/Z/16/Z, M.D.T.) and Wellcome Trust Discovery Award (WT225221/Z/22/Z, M.D.T. and L.V.W.). For the purpose of open access, the author has applied a CC BY public copyright licence to any Author Accepted Manuscript version arising from this submission. L.V.W. was supported by GSK/British Lung Foundation Chair in Respiratory Research. C.B. was supported by a UKRI Innovation Fellowship at Health Data Research UK (MR/S003762/1). C.J. held a Medical Research Council Clinical Research Training Fellowship (MR/P00167X/1). EXCEED is supported by the University of Leicester, the NIHR Leicester Respiratory Biomedical Research Centre, by Wellcome [202849, https://doi.org/10.35802/202849] and by Cohort Access fees from studies funded by the Medical Research Council (MRC), BBRSC, NIHR, the UK Space Agency, and GSK. It was previously supported by MRC grant G0902313. This work is supported by BREATHE - The Health Data Research Hub for Respiratory Health [UKR_PC_19004] in partnership with SAIL Databank. We also thank all participants and staff who have contributed their time to the study. Our analyses of UK Biobank and EXCEED used the ALICE and SPECTRE High Performance Computing Facilities at the University of Leicester. We want to acknowledge the participants of the Estonian Biobank for their contributions. The Estonian Biobank analyses were partially carried in the High Performance Computing Center, University of Tartu. Genes & Health is/has recently been core-funded by Wellcome (WT102627, WT210561), the Medical Research Council (UK) (M009017, MR/X009777/1, MR/X009920/1), Higher Education Funding Council for England Catalyst, Barts Charity (845/1796), Health Data Research UK (for London substantive site), and research delivery support from the NHS National Institute for Health Research Clinical Research Network (North Thames). Genes & Health is/has recently been funded by Alnylam Pharmaceuticals, Genomics PLC; and a Life Sciences Industry Consortium of Astra Zeneca PLC, Bristol-Myers Squibb Company, GlaxoSmithKline Research and Development Limited, Maze Therapeutics Inc, Merck Sharp & Dohme LLC, Novo Nordisk A/S, Pfizer Inc, Takeda Development Centre Americas Inc. We thank Social Action for Health, Centre of The Cell, members of our Community Advisory Group, and staff who have recruited and collected data from volunteers. We thank the NIHR National Biosample Centre (UK Biocentre), the Social Genetic & Developmental Psychiatry Centre (King's College London), Wellcome Sanger Institute, and Broad Institute for sample processing, genotyping, sequencing and variant annotation. We thank: Barts Health NHS Trust, NHS Clinical Commissioning Groups (City and Hackney, Waltham Forest, Tower Hamlets, Newham, Redbridge, Havering, Barking and Dagenham), East London NHS Foundation Trust, Bradford Teaching Hospitals NHS Foundation Trust, Public Health England (especially David Wyllie), Discovery Data Service/Endeavour Health Charitable Trust (especially David Stables), Voror Health Technologies Ltd (especially Sophie Don), NHS England (for what was NHS Digital) - for GDPR-compliant data sharing backed by individual written informed consent. Most of all we thank all of the volunteers participating in Genes & Health. We would like to thank the ThyroidOmics consortium, Nord-Trøndelag Health (HUNT) Study, and Michigan Genomics Initiative for depositing genome-wide summary statistics from previous studies in GWAS Catalog, and we are grateful to the staff who develop and maintain the GWAS Catalog.

## Author contributions

C.J. and M.D.T. supervised the study. A.T.W., J.C., N.S., M.D.T., and C.J. designed the study. A.T.W., J.C., K.C., C.Batini, A.I., R.P., E.A., S.K., and N.S. performed statistical analyses. A.T.W., J.C., K.C., C.Batini, A.I., R.P., E.A., S.K., E.J.H., W.H., B.S.R., F.D., L.V.W., N.S., M.D.T., C.J. analysed and/or interpreted the data. F.D. and L.V.W. provided methodological and statistical advice. A.T.W., J.C., N.S., M.D.T., and C.J. wrote the manuscript. D.J.S., R.C.F., N.J.B., I.N., N.R., C.E.B., L.V., E.A., C.Bee, S.E.W., M.P., A.L.H., T.E., D.S., B.M.J., D.A.H., Estonian Biobank Research Team and Genes & Health Research Team contributed to data collection, management or analysis of the EXCEED study, the Estonian Biobank or Genes & Health. All co-authors critically reviewed the manuscript.

## Competing interests

M.D.T. and L.V.W. have previously received funding from GSK for collaborative research projects outside of the submitted work. R.J.P., M.D.T., C.J., and L.V.W. have a funded research collaboration with Orion for collaborative research projects outside the submitted work. The remaining authors declare no competing interests.

## Additional information

[1]Department of Population Health Sciences, University of Leicester, Leicester, UK. [2]University Hospitals of Leicester NHS Trust, Infirmary Square, Leicester, UK. [3]Estonian Genome Center, Institute of Genomics, University of Tartu, Tartu, Estonia. [4]William Harvey Research Institute, Barts and the London School of Medicine and Dentistry, Queen Mary University of London, London, UK. [5]School of Computing and Mathematical Sciences, University of Leicester, Leicester, UK. [6]Department of Genetics and Genome Biology, University of Leicester, Leicester, UK. [7]Department of Cardiovascular Sciences, University of Leicester, Leicester, UK. [8]Division of Population Medicine, School of Medicine, Cardiff University, Cardiff, UK. [9]Institute for Lung Health, Leicester NIHR BRC, University of Leicester, Leicester, UK. [10]Department of Respiratory Sciences, University of Leicester, Leicester, UK. [11]Wolfson Institute of Population Health, Queen Mary University of London, London, UK. [12]Preventive Neurology Unit, Wolfson Institute of Population Health, Queen Mary University of London, London, UK. [13]Department of Neurology, Royal London Hospital, Barts Health NHS Trust, London, UK. [14]Blizard Institute, Barts and the London School of Medicine and Dentistry, Queen Mary University of London, London, UK. [15]Orion Pharma, Espoo, Finland. [16]Neuroscience Center, HiLIFE, University of Helsinki, Helsinki, Finland. [17]Institute for Molecular Medicine FIMM, HiLIFE, University of Helsinki, Helsinki, Finland. [18]Orion Pharma, Nottingham, UK. [19]These authors contributed equally: Alexander T. Williams, Jing Chen. [20]These authors jointly supervised this work: Martin D. Tobin, Catherine John. ✉e-mail: atw20@leicester.ac.uk; cj153@leicester.ac.uk

## Estonian Biobank Research Team

**Tõnu Esko**[3]

## Genes & Health Research Team

**Stavroula Kanoni** ⓘ [4], **Daniel Stow** ⓘ [11], **Benjamin M. Jacobs** ⓘ [12,13] **& David A. van Heel** ⓘ [14]

A full list of members and their affiliations appears in the Supplementary Information.

