## [Peer Review File · Nature Communications]

Genome-wide association study of thyroid-stimulating hormone highlights new genes, pathways and associations with thyroid diseaseREVIEWER COMMENTS

Reviewer #1 (Remarks to the Author):

Dr. Williams and colleagues, present a genome-wide association scan (GWAS) of the thyroid-stimulating hormone (TSH) involving more than 200,000 individuals from large biobanks or studies such as the Estonian Biobank (EstBB), UK Biobank (UKBB), and EXCEED, and integrating previously published GWAS results (Zhou et al, 2020). By doubling the sample size compared to previous efforts, the manuscript presents several new results in the field. The gene prioritization effort, the deeper assessment of thyroid-related phenotypes, and the derivation of polygenic scores (PGS) expand the results in a very meaningful way. However, there is a number of methodological issues that I would like to bring to the authors' attention. In general, I would highlight that the field of genetic epidemiology does not need larger numbers of findings but an assessment of the robustness and biological and clinical significance of the identified results, whether they are few or many. I would encourage the Authors to think that being more stringent with any significance criterion and presenting fewer but more relevant results can be a strength, not a limitation.

1) Overall, the study design is unclear. As far as I understood, GWAS were conducted on UKBB and EXCEED, summary statistics were pooled into a GWAS meta-analysis, then further pooled with GWAS summary statistics by Zhou et al 2020, with EstBB used only for locus validation. I would find more logical to pool the four GWAS together (UKBB, EXCEED, EstBB, and Zhou et al.) using appropriate meta-analytic approaches. This would provide greater discovery power than using arbitrary consistency criteria to expand the number of replicated SNPs (see Skol et al, Joint analysis is more efficient than replication-based analysis in GWAS, Nat Genet 2006). In addition, I'd recommend the Authors to outline the study design graphically at the beginning of the results. In the same section, especially given the number of studies is minimal, descriptive characteristics of involved studies (N, age, sex, % with hypo/hyperthyroidism, smoking, and other relevant characteristics) should be provided to allow the reader understand the context, particularly given TSH is strongly age-, sex-, and environment-related.

2) In UKBB and EXCEED, the Authors applied inverse normal transformation (INT) to the residuals from linear regression of TSH against age and sex. Then they conducted GWAS adjusting a second time for age and sex, in addition to the first 10 genetic principal components (PCs). Is this double adjustment for age and sex justified in some way? Please, compare with McCaw et al, Biometrics 2020; 76(4): 1262-1272, for the appropriateness of such an approach. Moreover, is this phenotype manipulation consistent with results by Zhou et al and EstBB, so that the meta-analysis is appropriate?

3) Why using a double standard for genetic variant filtering in UKBB and EXCEED? A minor allele count (MAC) >20 filter would be independent of the sample size. In fact, it would be more relevant to apply a MAC filter to EXCEED than to UKBB. The combination of MAC>20 in UKBB and minor allele frequency (MAF) >10% in EXCEED seems arbitrary and it is unclear how many SNPs were selected in each study and so, how much the results are entirely driven by UKBB.

4) The meta-analysis method is not described. The Authors indicate that they used Metal, but the software enables running different types of meta-analysis. Please, specify.

5) The criteria to identify significant loci are not clear. The Authors begin the Results section with "Using annotation informed fine-mapping (Online Methods), and a genome-wide significance threshold of $P < 5 \times 10^{-8}$, ...". This sounds like a $P < 5e-08$ was not the only criterion for selecting significant loci. However, I could not find in the Methods how and which annotation-informed fine-mapping was used to integrate the loci with $P < 5e-08$ in order to claim significance. Furthermore, additional, non-standard criteria involving P-values of 0.01 and 0.05 accompanied by request of direction consistency are introduced. At this stage, the pooling of EstBB data is described unclearly. I'd strongly recommend to stick to the $P < 5e-08$ as the only criterion to identify significant loci, maybe expanding the discovery sample size as suggested in point #1, which would also allow at least a vague quantification of the between-study heterogeneity.

6) In the formula to estimate the variance explained, I am not sure one can assume that the denominator V was always equal to 1, because INT was applied to regression residuals that were additionally adjusted for sex, age and 10 PCs in the GWAS process. However, the Authors have all information to derive the precise estimate of V . Please, verify, show, and adapt the variance explained estimates.

7) The term "signal" is used very extensively but not explained. It is very hard for a reader to understand what is a signal, a locus, an independent variant, and so on. Please, define and drop unnecessary genetic epidemiology slang (e.g.: sometimes, saying 'we identified XXX associated variants at XX loci' might be more easily understandable than 'we identified XXX signals').

8) The analysis of attenuation of the genetic associations when adjusting for smoking is very interesting but presented in an approximate way. If the Authors meant to conduct a mediation analysis, please set up an appropriate mediation analysis framework (there are many outlined in the literature). In addition, given that smoking definition is strongly dependent on the assessment method, please describe the smoking measurement method (questionnaire? self-admin? interviewer-admin?) and the smoking variable (never, former, current smokers?), and how each category was defined (e.g. how many months from quitting to define a former smoker? how many cigarettes per day to define an ever smoker?). Please, add this variable to the descriptive table discussed in point #1.

9) TSH was measured in a subset of UKBB and EXCEED participants. I'd invite the Authors to at least discuss the selection bias problem.

10) Please, revise the phenotype definition section in the Methods. For instance, the sentence "In all other instances, we were unable to disentangle true 0 measurements from those that may have arisen due to, for example, an individual's TSH being below the detectable range of the test apparatus used and, therefore, being entered into the primary care data as 0..." is quite unclear. In the end, given individuals with $TSH < 0.4$ were excluded anyway, the fact that TSH was truly = 0 or it was $<$ assay limit of detection of 0.05 seems irrelevant, correct?

11) Please, revise the section "Epidemiological associations ..." in the Methods. First, what is an "epidemiological association"? I think the term is misleading and can be removed. Why testing associations with T4 *OR* five clinical traits? Do the Authors mean "and"?

12) The Authors limited their analyses to TSH levels >0.4 and <4 . How do they think this might have limited the transportability of genetic loci and PGS associated with hypo- and hyperthyroidism? Wouldn't a complete uncensored analysis of all TSH levels have allowed identifying variants more relevant to the disease phenotypes and so the estimation of more meaningful PGS? I am asking this question especially in light of the INT, which would allow attenuation of spurious results on the tails of a distribution while preserving power to detect rare variant associations with disease statuses.

13) The "Identification of putative causal genes and causal variants" is an interesting analysis. However, the use of window sizes of ± 500 kb seems quite generous especially given evidence by Backman et al. (<https://www.nature.com/articles/s41586-021-04103-z>) showing that most of the times, the causal gene is the closest gene. Furthermore, using a validation criterion of ≥ 2 criteria to validate a candidate gene can be accepted but please, tone down claims such as "confidently implicated 112 priority genes" (conclusions, and throughout the manuscript) because ≥ 2 depends on how many criteria are set (eg: there would be more genes if the number of criteria was 10 or 15).

14) Please, discuss the limitations of the PGS and tone down claims. To be usable and transportable, PGS needs to be calibrated and their discrimination ability needs to be tested in independent settings. For the same reason, please use more prudential wording in the conclusions and abstract.

15) Regarding PGS: can the Authors really claim that the PGS predict the age of onset of hypo- and hyperthyroidism? ie: does the PGS reliably estimates at which age one has the disease onset,

or would the PGS predict the probability of a person to get the disease by a certain age? In addition, please, don't use terms such as "strongly predicts" but let the reader understand what this means (ie: clearly discriminate between those who may and may not develop a disease). Please, check the wording, especially in the Abstract.

16) In the Abstract, Introduction and Discussion, please remove claims of primacy and celebration and use the space to discuss findings in greater details, list strengths and limitations of the current study, etc.

17) Define which specific imputation score was used in variant filtering in UKBB and EXCEED.

Reviewer #2 (Remarks to the Author):

The authors conducted a GWAS on thyroid-stimulating hormone (TSH) levels based on UK Biobank and meta-analysed their results with previous GWAS from Zhou et al and the Estonian Biobank). They detected 260 independent signals for TSH at 156 loci, of which 158 signals at 78 loci were new. Subsequently, they have fine-mapped these susceptibility regions to highlight putative causal variants and then integrated 8 criteria to undertake a variant-to-gene mapping to identify 112 putative causal genes for TSH levels satisfying at least 2 criteria. They also performed several analyses to reveal functional association : association between the 260 TSH-associated variants and other diseases, pathways enrichment analysis for the 112 genes, association between pathway specific TSH genetic risk scores (for 26 enriched pathways) and other diseases, association between polygenic score (PGS) for TSH and other diseases. They also show that the PGS for TSH was associated to early onset hypo- and hyperthyroidism.

General comments :

This GWAS is based on a large sample size of individuals (n=247 000). The authors reported new susceptibility loci and conducted a comprehensive analysis to identify the genes and pathways involved in the TSH levels.

However, major improvements are needed to make the paper clearer. Some important informations/definitions are missing or sometimes not located at the good place in the manuscript. In particular, the authors should be more explicit in the definitions of some terms (such as sentinel SNP, 95% credible set, etc.) or in the decisions they took (particularly in the definition of phenotypes), and also justify or explain their methodology. There is in general a lack of description of the studied phenotypes. A major limitation of this GWAS analysis, that is also discussed in the discussion part, is the lack of replication steps for the new identified signals. The authors also explain that they perform prediction performance for the PGS analysis, but from my understanding only association tests were performed. I think that it would have been actually useful to formally test the prediction performance of the PGS for TSH levels.

More specifically :

- Line 82 : The studies included in the GWAS is not clear. From my understanding of the Supp Table 1, a meta-analysis of EXCEED, UKBB, Est BB and study from Zhou et al. was conducted. However only consistency of effect from UKBB, Est BB and study from Zhou et al. was checked. Also, in line 82, it is stated that the GWAS was conducted in EXCEED and UKBB and then meta-analysed with Zhou et al., but in line 88 it is stated that the meta-analysis included UKBB, Est BB and Zhou et al. There is a need for clarification.

Also, the EXCEED, UKBB, Est BB studies are described in the Supplementary Note, but not the study from Zhou et al. This study should also be detailed (which population, number of individuals etc.) in the Supplementary Note.

- Lines 83-94 : It would have been useful to show a manhattan plot to localise the different susceptibility loci and their significance.

The supplementary Table 1 is confusing. It may be clearer to show on the first columns, the SNPs with the lowest p-values from the GWAS and then the sentinel SNP and show the results from the

8 different methods for variant-to-gene mapping as explained in lines 95-105. Here, the table seems to mixed all these results, making it difficult to read and follow. Also, the authors should consider to show the name of the locus to make the comparison with results from other GWAS easier (this should also be considered in Supplementary tables 14 and 16).

- Line 95 : An explanation on the definition of the sentinel variant is missing here before explaining the variant-to-gene mapping

- Lines 96-104 : For integration of eight sources of evidence for identification of putative causal genes, the authors considered genes satisfying ≥ 2 criteria for subsequent analyses. It may be useful to discuss this choice in the discussion as a limit. I do not think all eight criteria have the same biological effect, especially in terms of functional consequences. It could have been better to define a new concept by applying proper weight to each criterion (based on their functional effect) to calculate a score for each gene.

- Lines 106-108 : the word « comparison » is not clear here. I guess what the authors meant was instead that among the 112 putative causal genes identified, 67 were previously reported.

In Figure 1, only 7 criteria were shown while the number of criteria in the manuscript was 8.

The information from supplementary Table 3 should be directly added to supplementary Table 2 for more clarity (and then removed Supp Table 3).

- Line 109 : the authors mentioned that "78 genes have not been previously implicated in TSH level". I found this number as "76" based on the "novel" column (based on TRUE entries) and "77" based on the "n_novel_signals" column (values greater than 0) in the Supplementary Table S1. Which one is correct?

- Lines 123-125 : It may be clearer to add here that the association test was done for the 260 sentinel SNPs.
Also in Supplementary Table 4, it is written "257 top PIP variants" while 260 variants are tested.

- Line 126-128 : On how many variants was the PheWAS analysis performed ? It should be mentioned here on which datasets these analyses were done (UKBB only ? Or UKBB + EXCEED ? Was EXCEED considered as part of UKBB ?). Supplementary Figures 1 are not easy to read, it may be useful to find a more synthetic way to present these results, or at least it may help to comment these results further in the text.

- Information of supplementary Figure 2 could have been merged to Figure 1 to reduce the number of Figures.

- In supplementary Figure 2 and supplementary Table 6, « hypo_due_to_Rx » needs to be explained in the footnotes

- Line 165 : the authors highlighted 3 genes but describe only the function of APOH and did not comment on the findings on SPATA6 and ADCY6.

- Line 242 : Titles for Figure 3 and Figure 4 should remove the words « Prediction performance » as from my understanding the prediction performance of the PGS is not evaluated in the paper but these figures only show the association between PGS and different traits, or PGS and TSH in population of different ancestries. Why not formally evaluate the prediction (for instance using ROC curve and AUC) ?

- Line 283 : the sensitivity analyses on hypo and hyperthyroidism excluded individuals who were included in the TSH GWAS. How come those individuals were diagnosed for thyroid disorders without dosage of TSH levels ? Is there many individuals concerned by this ?

- Line 329 : the predictive power was not evaluated.

- Lines 348-350 : the classification of thyroid diseases were expected to be described in the methods part, not in the discussion. I think that in the discussion, limitation on the definition of TSH levels in UKBB and EXCEED should be mentioned as well as in the definition of thyroid disorders in UKBB. For instance, were treatment for thyroid diseases could be taken into account ?

- Lines 372-383 :

The authors stated that they removed individuals with first TSH measurement equal to 0, except if the individuals also reported hyperthyroidism. But then, they stated that individuals with TSH<0.4 were removed anyway. It seems that this paragraph could be simplified as individuals with a value of 0 are removed regardless of their hyperthyroidism status.

The exact number of individuals from UKBB and EXCEED with TSH levels available and those included in the analyses should be mentioned. Maybe a flow chart may be useful to show the number of individuals that were removed at each step of the quality controls (TSH levels, relatedness, etc). Some descriptive statistics of the phenotypes such as mean values of TSH, distribution of age at time measurement and proportion of men/women are missing.

- Line 383 :

The TSH intervals [0.4-4.0] should be explained. I do not think that this is sufficient to say that was done previously. The normal ranges of TSH levels depend on the age distribution of the population, and TSH levels tend to be higher in older population (Surks MI, Boucai L. Age- and race-based serum thyrotropin reference limits. J Clin Endocrinol Metab 2010;95(2):496-502).

- Line 386 : the authors need to explain why they did the genome-wide association analysis in two steps by deriving the residuals of the linear regression of TSH against age and sex in a first step and then a GWAS analysis on the inverse normal transformation of these residuals. Why TSH concentrations were not directly analysed in the GWAS ? It seems that the GWAS was done differently in Zhou et al and the Est BB study.

- Line 400 : It should be precised whether the meta-analysis was done using a fixed effect or a random meta-analysis? Has a heterogeneity test between studies been performed?

- Line 404 : what was the estimation of the LD Score regression intercept ?

- Line 405 : sentinel SNP should be defined in the text and not only in the legend of tables

- Line 423 : « 95% credible set » need to be defined.

- Lines 450-453 : the definition of hypo-, hyperthyroidism, other thyroid diseases from UKBB data is not clear. These phenotypes are usually tricky to defined and the clinical codes presented in sup Table 15 are not sufficient, the authors should be more explicit on the variables used and how they were combined them to define those diseases. What were the « other thyroid diseases » ? For instance, it is stated in line 146 that secondary hypothyroidism were excluded from the definition of hypothyroidism. This should be detailed in this methodological part.

From my understanding of the paragraph lines 146-152, treated hyperthyroidism were analyzed separately, it is not clear why these cases were not considered as hyperthyroidism directly.

Also descriptive statistics on the number of cases and controls used on the analysis of hypo-, hyperthyroidism, thyroid cancer and other thyroid disease, as well as their distribution by age, sex.

Also, I suggest to put the supplementary table 15 in the excel file with all other supplementary tables for more clarity.

- Line 455 : I suppose that a logistic regression was used, this should be mentioned. How were the controls defined ? Did you use the same group of controls for each thyroid diseases ?

- Line 458 : at what time was current smoking status or pack-years defined ? What was the proportion of smokers in the analysed samples, and the mean pack-years ?

- Line 501 : In the polygenic priority score (PoPS) analysis, why the authors considered window size of $\pm 250\text{kb}$ while in all other analyses in the manuscript a window size of $\pm 500\text{kb}$ was considered?

- In the Supplementary Figures file, after Figure S4, there is a "Supplementary note" that is the same than the Supplementary notes shown before Figure S1 and in the next page there is a Figure that is in the main paper.

- Reference #7 (DeepPheWAS) needs to be updated. It is now published in the "Bioinformatics" journal.

- Reference #15 (FinnGen) needs to be updated. It is now published in the "Nature" journal.

Response to Reviewers

Reviewer #1 (Remarks to the Author):

Dr. Williams and colleagues, present a genome-wide association scan (GWAS) of the thyroid-stimulating hormone (TSH) involving more than 200,000 individuals from large biobanks or studies such as the Estonian Biobank (EstBB), UK Biobank (UKBB), and EXCEED, and integrating previously published GWAS results (Zhou et al, 2020). By doubling the sample size compared to previous efforts, the manuscript presents several new results in the field. The gene prioritization effort, the deeper assessment of thyroid-related phenotypes, and the derivation of polygenic scores (PGS) expand the results in a very meaningful way. However, there is a number of methodological issues that I would like to bring to the authors' attention. In general, I would highlight that the field of genetic epidemiology does not need larger numbers of findings but an assessment of the robustness and biological and clinical significance of the identified results, whether they are few or many. I would encourage the Authors to think that being more stringent with any significance criterion and presenting fewer but more relevant results can be a strength, not a limitation.

Thank you. We agree with these general points and we address the specific concerns below.

1) Overall, the study design is unclear. As far as I understood, GWAS were conducted on UKBB and EXCEED, summary statistics were pooled into a GWAS meta-analysis, then further pooled with GWAS summary statistics by Zhou et al 2020, with EstBB used only for locus validation. I would find more logical to pool the four GWAS together (UKBB, EXCEED, EstBB, and Zhou et al.) using appropriate meta-analytic approaches. This would provide greater discovery power than using arbitrary consistency criteria to expand the number of replicated SNPs (see Skol et al, Joint analysis is more efficient than replication-based analysis in GWAS, Nat Genet 2006). In addition, I'd recommend the Authors to outline the study design graphically at the beginning of the results. In the same section, especially given the number of studies is minimal, descriptive characteristics of involved studies (N, age, sex, % with hypo/hyperthyroidism, smoking, and other relevant characteristics) should be provided to allow the reader understand the context, particularly given TSH is strongly age-, sex-, and environment-related.

Thank you for this constructive feedback. We have amended our design such that the existing, single meta-analysis of UK Biobank, EXCEED and Zhou et al (Stage 1) is then meta-analysed with the Estonian Biobank and an additional study (Genes & Health) (Stage 2). We agree that this provides a more powerful approach than replication.

We provide a new figure (Figure 1) as suggested to provide greater clarity for the reader, and additionally include descriptive characteristics of the cohorts in a new Supplementary Table 1.

2) In UKBB and EXCEED, the Authors applied inverse normal transformation (INT) to the residuals from linear regression of TSH against age and sex. Then they conducted GWAS adjusting a second time for age and sex, in addition to the first 10 genetic principal components (PCs). Is this double adjustment for age and sex justified in some way? Please, compare with McCaw et al, Biometrics 2020; 76(4): 1262-1272, for the appropriateness of such an approach. Moreover, is this phenotype manipulation consistent with results by Zhou et al and EstBB, so that the meta-analysis is appropriate?

Both McCaw et al (2020) and Sofer et al (2019, DOI: 10.1002/gepi.22188) argue for the inclusion of all covariates in the "second stage" regression (i.e. the association analysis) as

well as the “first stage” (regression of the phenotype on covariates to obtain residuals). We used this approach in our analysis of UK Biobank and EXCEED, and the same approach has been used in all other studies included in our analysis. Thank you for highlighting that this was not clear; we have added the following clarification in the Methods: “All included studies used the same approach to covariate adjustment as applied in stage 1.” (third paragraph of “Stage 2 analysis” section, page 12)

3) Why using a double standard for genetic variant filtering in UKBB and EXCEED? A minor allele count (MAC) >20 filter would be independent of the sample size. In fact, it would be more relevant to apply a MAC filter to EXCEED than to UKBB. The combination of MAC>20 in UKBB and minor allele frequency (MAF) >10% in EXCEED seems arbitrary and it is unclear how many SNPs were selected in each study and so, how much the results are entirely driven by UKBB.

Thank you for highlighting the lack of clarity around variant filtering in UK Biobank and EXCEED. We used a minor allele frequency of 0.1% (not 10%) in EXCEED which equates to a minor allele count of around 6. While this is a less stringent threshold than was used in our UK Biobank analysis, only one of our 260 sentinel variants (rs141336511) had a minor allele count less than 20 in EXCEED. Since this variant was also captured in UK Biobank and Zhou et al, it is unlikely to have had an impact on our findings had we used a stricter minor allele count threshold for EXCEED. We have added the number of variants tested in UK Biobank and EXCEED to the Supplementary Note and the new Figure 1.

4) The meta-analysis method is not described. The Authors indicate that they used Metal, but the software enables running different types of meta-analysis. Please, specify.

We have added some text to the third paragraph of “Stage 1 analysis” (Methods, page 11) to clarify our approach.

5) The criteria to identify significant loci are not clear. The Authors begin the Results section with “Using annotation informed fine-mapping (Online Methods), and a genome-wide significance threshold of $P < 5 \times 10^{-8}$, ...”. This sounds like a $P < 5e-08$ was not the only criterion for selecting significant loci. However, I could not find in the Methods how and which annotation-informed fine-mapping was used to integrate the loci with $P < 5e-08$ in order to claim significance. Furthermore, additional, non-standard criteria involving P-values of 0.01 and 0.05 accompanied by request of direction consistency are introduced. At this stage, the pooling of EstBB data is described unclearly. I’d strongly recommend to stick to the $P < 5e-08$ as the only criterion to identify significant loci, maybe expanding the discovery sample size as suggested in point #1, which would also allow at least a vague quantification of the between-study heterogeneity.

Thank you for highlighting that the above text in the Results was unclear. $P < 5 \times 10^{-8}$ was the only criterion for defining significance. We have simplified the text of the Results and refer the reader to the Online Methods for details. In the Online Methods under the heading “Sentinel variant selection and fine mapping” (page 12) we specify “Using Stage 1 results only, we selected 2Mb loci centred on the most significant variant for all regions containing a variant with $P < 5 \times 10^{-8}$ ” and provide further details of signal selection and fine mapping.

Additionally, we expanded the sample size in Stage 2, as described in the new Figure 1 described in the response to point (1) above.

6) In the formula to estimate the variance explained, I am not sure one can assume that the denominator V was always equal to 1, because INT was applied to regression residuals that were

additionally adjusted for sex, age and 10 PCs in the GWAS process. However, the Authors have all information to derive the precise estimate of V. Please, verify, show, and adapt the variance explained estimates.

Thank you. After the additional adjustment, the estimate of V was 0.9916, which changes the estimate of variance explained from 22.6% to 22.8%. This has been updated accordingly in the Results (second paragraph of "TSH association with 260 sentinel variants", page 3). We have also revised our approach to heritability, using an independent estimate of heritability from the largest twin study (by Panicker et al.) to calculate the proportion of heritability explained by our sentinels.

7) The term "signal" is used very extensively but not explained. It is very hard for a reader to understand what is a signal, a locus, an independent variant, and so on. Please, define and drop unnecessary genetic epidemiology slang (e.g.: sometimes, saying 'we identified XXX associated variants at XX loci' might be more easily understandable than 'we identified XXX signals').

Thank you for highlighting this. We have removed references to "signals" throughout the text and instead focused on sentinel variants (defined in the first paragraph of "Sentinel variant selection and fine mapping" in Methods, page 12) and loci.

8) The analysis of attenuation of the genetic associations when adjusting for smoking is very interesting but presented in an approximate way. If the Authors meant to conduct a mediation analysis, please set up an appropriate mediation analysis framework (there are many outlined in the literature). In addition, given that smoking definition is strongly dependent on the assessment method, please describe the smoking measurement method (questionnaire? self-admin? interviewer-admin?) and the smoking variable (never, former, current smokers?), and how each category was defined (e.g. how many months from quitting to define a former smoker? how many cigarettes per day to define an ever smoker?). Please, add this variable to the descriptive table discussed in point #1.

Thank you for highlighting this. We did not intend to conduct a formal mediation analysis, but at the time of submission we were unable to undertake a lookup of associations with smoking behaviour of TSH-associated sentinels in the latest and largest GWAS of smoking behaviour as the GSCAN consortium had not yet released the results. After contacting the GSCAN consortium, we have been able to complete such lookups and therefore we have replaced the smoking-adjusted analysis with a look-up of the 260 TSH sentinel variants in summary statistics from GSCAN across four smoking behaviour phenotypes. This is a more powerful approach given the sample size of GSCAN.

As a result of this change we have not used smoking variables in our own analysis, but for completeness we have added a descriptor of smoking behaviour to the new Supplementary Table 1, and ensured this is defined in the table legend.

9) TSH was measured in a subset of UKBB and EXCEED participants. I'd invite the Authors to at least discuss the selection bias problem.

Thank you for highlighting this important consideration. We have now included a discussion of possible selection bias in the Discussion (page 10).

10) Please, revise the phenotype definition section in the Methods. For instance, the sentence "In all other instances, we were unable to disentangle true 0 measurements from those that may have arisen due to, for example, an individual's TSH being below the detectable range of the test

apparatus used and, therefore, being entered into the primary care data as 0..." is quite unclear. In the end, given individuals with TSH < 0.4 were excluded anyway, the fact that TSH was truly = 0 or it was < assay limit of detection of 0.05 seems irrelevant, correct?

We agree, and have simplified our description of the phenotype accordingly. ("Phenotype" (Methods), page 11)

*11) Please, revise the section "Epidemiological associations ..." in the Methods. First, what is an "epidemiological association"? I think the term is misleading and can be removed. Why testing associations with T4 *OR* five clinical traits? Do the Authors mean "and"?*

We have removed the word "epidemiological" from the corresponding section header and updated Supplementary Table 5 to reflect this change.

We have separated the sentence relating to associations with T4 and clinical phenotypes into two sentences (page 13, paragraph 3). We hope this improves understanding.

12) The Authors limited their analyses to TSH levels >0.4 and <4. How do they think this might have limited the transportability of genetic loci and PGS associated with hypo- and hyperthyroidism? Wouldn't a complete uncensored analysis of all TSH levels have allowed identifying variants more relevant to the disease phenotypes and so the estimation of more meaningful PGS? I am asking this question especially in light of the INT, which would allow attenuation of spurious results on the tails of a distribution while preserving power to detect rare variant associations with disease statuses.

Thank you. We now include mention of this potential limitation in the Discussion.

We followed the approach in a previous GWAS of TSH by Teumer et al. The ThyroidOmics consortium, which contributed around 45% of the participants in the meta-analysis by Zhou et al., also took this approach, whilst HUNT (46% of participants in the Zhou et al. analysis) employed an unrestricted main analysis, but a sensitivity analysis restricted to TSH measures within the normal range found similar results. For the 260 sentinel variants, we compared the $-\log_{10}P$ values from association analyses using the whole range of TSH measures (unrestricted) vs using the restricted range (figure below) and found generally higher p-values for the unrestricted range. Whilst these findings suggest that we did not lose power from the approach we adopted, we accept that there may be other disadvantages to excluding extreme values, and we have added the following to the discussion "Our approach of excluding measurements of TSH outside the normal range could miss some variants more relevant to the extremes of the distribution" (page 10, paragraph 1).

13) The “Identification of putative causal genes and causal variants” is an interesting analysis. However, the use of window sizes of +/-500kb seems quite generous especially given evidence by Backman et al. (<https://www.nature.com/articles/s41586-021-04103-z>) showing that most of the times, the causal gene is the closest gene. Furthermore, using a validation criterion of >=2 criteria to validate a candidate gene can be accepted but please, tone down claims such as “confidently implicated 112 priority genes” (conclusions, and throughout the manuscript) because >=2 depends on how many criteria are set (eg: there would be more genes if the number of criteria was 10 or 15).

We agree that recent evidence suggests that the nearest gene is often – but not always – the causal gene. Of our 112 prioritised genes, 84 genes were implicated (at least partly) by a distance-based criterion within a +/-500kb window; however, for 68 (81%) of these the implicated gene was also the nearest gene to the sentinel. Of the remaining 16 that were not the nearest gene, 15 were nevertheless within a +/-250kb window. This is consistent with evidence from Backman et al. and others.

We have adopted a more comprehensive approach for reporting implicated genes than in previous studies, where a single criterion, e.g. nearest gene or eQTL colocalization, has often been used. We have removed the term “high confidence gene” and discuss limitations of our variant-to-gene mapping approach in the Discussion (page 10, paragraph 2).

14) Please, discuss the limitations of the PGS and tone down claims. To be usable and transportable, PGS needs to be calibrated and their discrimination ability needs to be tested in independent settings. For the same reason, please use more prudential wording in the conclusions and abstract.

Thank you for bringing this to our attention. We have revised the wording of the manuscript to ensure that the association analyses with the PGS are appropriately described, and have also rephrased the abstract accordingly. We have also revised the claims in the discussion (pages 8 and 9).

To further evaluate the PGS, we have produced receiver operating characteristic (ROC) curves which are now included as Supplementary Figure 6 and we describe the area under the curve (AUC) in the manuscript. We have additionally conducted association testing using data from an independent South Asian population sourced from Genes & Health study (Results and Supplementary Table 12).

15) Regarding PGS: can the Authors really claim that the PGS predict the age of onset of hypo- and hyperthyroidism? ie: does the PGS reliably estimates at which age one has the disease onset, or would the PGS predict the probability of a person to get the disease by a certain age? In addition, please, don't use terms such as “strongly predicts” but let the reader understand what this means (ie: clearly discriminate between those who may and may not develop a disease). Please, check the wording, especially in the Abstract.

We have revised the wording throughout the abstract and paper to refer to our results as associations.

16) In the Abstract, Introduction and Discussion, please remove claims of primacy and celebration and use the space to discuss findings in greater details, list strengths and limitations of the current study, etc.

Thank you. We have removed the phrases “the largest” and “for the first time” from the abstract. We retained mention of the utility of EHR in increasing the available sample size in the Abstract and Discussion, to make it clear how our study adds to the existing literature.

In the Discussion section, we have extended the discussion of the limitations of our study (page 10).

17) Define which specific imputation score was used in variant filtering in UKBB and EXCEED.

This information is included in the Supplementary Note (page 2, paragraph 6 for EXCEED, and page 3, paragraph 1 for UKB).

Reviewer #2 (Remarks to the Author):

The authors conducted a GWAS on thyroid-stimulating hormone (TSH) levels based on UK Biobank and meta-analysed their results with previous GWAS from Zhou et al and the Estonian Biobank). They detected 260 independent signals for TSH at 156 loci, of which 158 signals at 78 loci were new. Subsequently, they have fine-mapped these susceptibility regions to highlight putative causal variants and then integrated 8 criteria to undertake a variant-to-gene mapping to identify 112

putative causal genes for TSH levels satisfying at least 2 criteria. They also performed several analyses to reveal functional association : association between the 260 TSH-associated variants and other diseases, pathways enrichment analysis for the 112 genes, association between pathway specific TSH genetic risk scores (for 26 enriched pathways) and other diseases, association between polygenic score (PGS) for TSH and other diseases. They also show that the PGS for TSH was associated to early onset hypo- and hyperthyroidism.

General comments :

This GWAS is based on a large sample size of individuals (n=247 000). The authors reported new susceptibility loci and conducted a comprehensive analysis to identify the genes and pathways involved in the TSH levels.

However, major improvements are needed to make the paper clearer. Some important informations/definitions are missing or sometimes not located at the good place in the manuscript. In particular, the authors should be more explicit in the definitions of some terms (such as sentinel SNP, 95% credible set, etc.) or in the decisions they took (particularly in the definition of phenotypes), and also justify or explain their methodology. There is in general a lack of description of the studied phenotypes. A major limitation of this GWAS analysis, that is also discussed in the discussion part, is the lack of replication steps for the new identified signals. The authors also explain that they perform prediction performance for the PGS analysis, but from my understanding only association tests were performed. I think that it would have been actually useful to formally test the prediction performance of the PGS for TSH levels.

Thank you for these helpful comments. We have addressed the issues regarding clarity of study design and phenotype definition, and have additionally revised our overall study design to a two-stage design in line with comments from Reviewer #1. We have also improved our approach to issues of predictive performance. Please see our responses below for further specific detail on each point.

More specifically :

- Line 82 : The studies included in the GWAS is not clear. From my understanding of the Supp Table 1, a meta-analysis of EXCEED, UKBB, Est BB and study from Zhou et al. was conducted. However only consistency of effect from UKBB, Est BB and study from Zhou et al. was checked. Also, in line 82, it is stated that the GWAS was conducted in EXCEED and UKBB and then meta-analysed with Zhou et al., but in line 88 it is stated that the meta-analysis included UKBB, Est BB and Zhou et al. There is a need for clarification.

Thank you for highlighting this. We undertook separate GWAS in EXCEED and UK Biobank, for which we had access to individual-level data. We meta-analysed both set of results with publicly-available summary statistics from Zhou et al. We have amended the wording in the first paragraph of the Results to reflect this, and have added a new study design figure (Figure 1) and restructured Supplementary Table 2. We hope this clarifies our approach.

Also, the EXCEED, UKBB, Est BB studies are described in the Supplementary Note, but not the study from Zhou et al. This study should also be detailed (which population, number of individuals etc.) in the Supplementary Note.

We have added text to the Supplementary Note (page 3, paragraph 2) to describe the study by Zhou et al. We have also added a new supplementary table (Supplementary Table 1) to describe demographics of individuals included in our study.

- Lines 83-94 : It would have been useful to show a manhattan plot to localise the different susceptibility loci and their significance. The supplementary Table 1 is confusing. It may be clearer to show on the first columns, the SNPs with the lowest p-values from the GWAS and then the sentinel SNP and show the results from the 8 different methods for variant-to-gene mapping as explained in lines 95-105. Here, the table seems to mixed all these results, making it difficult to read and follow. Also, the authors should consider to show the name of the locus to make the comparison with results from other GWAS easier (this should also be considered in Supplementary tables 14 and 16).

Thank you for these comments. We have now included a Manhattan plot as suggested (Supplementary Figure 1). To aid readability, we have restructured and added a more comprehensive legend to Supplementary Table 1 (in the original version; now Supplementary Table 2 in the revised version). We have also improved the legend in Supplementary Table 2 (now Supplementary Table 3).

- Line 95 : An explanation on the definition of the sentinel variant is missing here before explaining the variant-to-gene mapping

We have added a brief description of “sentinel variant” to the second paragraph of “TSH association with 260 variants” (Results), with a fuller description in the Methods.

- Lines 96-104 : For integration of eight sources of evidence for identification of putative causal genes, the authors considered genes satisfying ≥ 2 criteria for subsequent analyses. It may be useful to discuss this choice in the discussion as a limit. I do not think all eight criteria have the same biological effect, especially in terms of functional consequences. It could have been better to define a new concept by applying proper weight to each criterion (based on their functional effect) to calculate a score for each gene.

We agree that weighting of the criteria would be a very helpful methodological development, but in planning our analysis we felt that without additional reference data, weighting would be arbitrary. Since conclusions could be sensitive to the choice of weighting, we adopted a simple tally of evidence, as used in another recent publication (PMID: 36474045). The requirement of at least two criteria to identify a putative causal gene is more stringent than previous approaches which often report putative causal genes implicated by one line of evidence. We have added some text to the discussion regarding this limitation (page 10, paragraph 2).

- Lines 106-108 : the word « comparison » is not clear here. I guess what the authors meant was instead that among the 112 putative causal genes identified, 67 were previously reported.

Thank you; we have edited the wording of this sentence to improve clarity.

In Figure 1, only 7 criteria were shown while the number of criteria in the manuscript was 8.

We had omitted the column relating to the pQTL associations as there were no significant findings. We have added a sentence to the Figure 2 legend to clarify this.

The information from supplementary Table 3 should be directly added to supplementary Table 2 for more clarity (and then removed Supp Table 3).

Supplementary Table 3 (now Supplementary Table 4) lists all previously reported genes, which do not fully overlap with our prioritised genes in Supplementary Table 2 (now Supplementary Table 3), so we have presented these separately for clarity.

- Line 109 : the authors mentioned that “78 genes have not been previously implicated in TSH level”. I found this number as “76” based on the “novel” column (based on TRUE entries) and “77” based on the “n_novel_signals” column (values greater than 0) in the Supplementary Table S1. Which one is correct?

Thank you for highlighting this. The correct number is 76. We have amended the text to reflect this. We have also ensured that there is only one column (“novel_signal”) in Supplementary Table 1 (as submitted, now Supplementary Table 2) which gives information on novelty of a signal.

- Lines 123-125 : It may be clearer to add here that the association test was done for the 260 sentinel SNPs.

Also in Supplementary Table 4, it is written “257 top PIP variants” while 260 variants are tested.

We have added text to the fourth paragraph of “Identification of putative causal genes and causal variants” (Results, page 3) to clarify that we only tested sentinel variants. We tested 257 sentinel variants with clinical thyroid disease and free T4 since three of our sentinel variants (all variants with minor allele frequency <1%) were not available in UK Biobank (where these association tests were performed). These three sentinel variants were available in the Zhou et al study only. We have clarified this in Supplementary Table 4 (which is now Supplementary Table 5).

- Line 126-128 : On how many variants was the PheWAS analysis performed ? It should be mentioned here on which datasets these analyses were done (UKBB only ? Or UKBB + EXCEED ? Was EXCEED considered as part of UKBB ?). Supplementary Figures 1 are not easy to read, it may be useful to find a more synthetic way to present these results, or at least it may help to comment these results further in the text.

We have added the number of sentinel variants tested in the PheWAS the fourth paragraph of “Identification of putative causal genes and causal variants” (Results, page 4). We have also clarified that this PheWAS was conducted in UK Biobank. The PheWAS results for the sentinel variants tested are also presented in Supplementary Table 6.

- Information of supplementary Figure 2 could have been merged to Figure 1 to reduce the number of Figures.

The additional phenotype included in Supplementary Figure 3 (labelled as hypothyroidism due to treatment) largely represents individuals with hyperthyroidism who have been treated, for example with medications, surgery or radioablation, though in some cases this treatment may have been for other conditions. We believe this phenotype may be of interest to some readers, but to retain flow and brevity in the main manuscript, we have retained the figure in the supplement.

- In supplementary Figure 2 and supplementary Table 6, « hypo_due_to_Rx » needs to be explained in the footnotes

Thank you for highlighting this. We have added text to explain this to the legends of Supplementary Figure 3 and Supplementary Table 7 (formerly Supplementary Figure 2 and Supplementary Table 6).

- Line 165 : the authors highlighted 3 genes but describe only the function of APOH and did not comment on the findings on SPATA6 and ADCY6.

Thank you for highlighting this. Given space limitations, we focused on the well-characterised *APOH*, but have now added some description of *SPATA6*, *ADCY6* and relevant PheWAS findings. (page 5, paragraph 3)

- Line 242 : Titles for Figure 3 and Figure 4 should remove the words « Prediction performance » as from my understanding the prediction performance of the PGS is not evaluated in the paper but these figures only show the association between PGS and different traits, or PGS and TSH in population of different ancestries. Why not formally evaluate the prediction (for instance using ROC curve and AUC) ?

Thank you for highlighting this. We have revised the legends of Figures 3-5 accordingly (now Figures 4-6).

As requested, we have conducted further analysis to evaluate the prediction performance of our TSH PGS on different thyroid diseases in the UK Biobank European population. To do so, we utilised receiver operating characteristic (ROC) curves and calculated the area under the curve (AUC), now described in the Results (page 8, paragraph 2) and shown in Supplementary Figure 6.

- Line 283 : the sensitivity analyses on hypo and hyperthyroidism excluded individuals who were included in the TSH GWAS. How come those individuals were diagnosed for thyroid disorders without dosage of TSH levels ? Is there many individuals concerned by this ?

Hypo- and hyperthyroidism were defined using clinical codes reflecting diagnoses in primary care and secondary care as well as in self-reported data (from questionnaire and nurse interview). As TSH measurements were captured from laboratory results within the overall primary care record alone and primary care data is currently available for only half of UK Biobank, there were a large number of individuals who met our definitions of hypo- and hyperthyroidism (from secondary care and self-report) but do not have a TSH measurement that was available to us.

- Line 329 : the predictive power was not evaluated.

We have added this analysis into our manuscript to provide additional insight into the performance of our TSH PGS as suggested by the reviewer (also see the response above).

- Lines 348-350 : the classification of thyroid diseases were expected to be described in the methods part, not in the discussion. I think that in the discussion, limitation on the definition of TSH levels in UKBB and EXCEED should be mentioned as well as in the definition of thyroid disorders in UKBB. For instance, were treatment for thyroid diseases could be taken into account ?

We have added some further description of how thyroid diseases were defined in the Methods section (page 13, paragraph 3).

With regard to treatment for thyroid diseases, we took an individual's first non-missing TSH measurement to minimise the effect of thyroid function-altering medications on our phenotype as an individual is unlikely to have received these medications before their first thyroid function test. This is described in the Methods (page 10, paragraph 2). We have also acknowledged the potential effect of thyroid medications in the Discussion (page 10, paragraph 1).

- Lines 372-383 :

The authors stated that they removed individuals with first TSH measurement equal to 0, except if

the individuals also reported hyperthyroidism. But then, they stated that individuals with TSH<0.4 were removed anyway. It seems that this paragraph could be simplified as individuals with a value of 0 are removed regardless of their hyperthyroidism status.

The exact number of individuals from UKBB and EXCEED with TSH levels available and those included in the analyses should be mentioned. Maybe a flow chart may be useful to show the number of individuals that were removed at each step of the quality controls (TSH levels, relatedness, etc). Some descriptive statistics of the phenotypes such as mean values of TSH, distribution of age at time measurement and proportion of men/women are missing.

Thank you; we have simplified the description of the phenotype data management as suggested. We now include descriptive statistics, where they were available, in a new Supplementary Table 1.

- Line 383 :

The TSH intervals [0.4-4.0] should be explained. I do not think that this is sufficient to say that was done previously. The normal ranges of TSH levels depend on the age distribution of the population, and TSH levels tend to be higher in older population (Surks MI, Boucai L. Age- and race-based serum thyrotropin reference limits. J Clin Endocrinol Metab 2010;95(2):496-502).

We agree that the optimal cut-off of TSH values is not clear. We now include mention of this potential limitation in the Discussion. Please also see comments in response to Reviewer 1 (point 12).

- Line 386 : the authors need to explain why they did the genome-wide association analysis in two steps by deriving the residuals of the linear regression of TSH against age and sex in a first step and then a GWAS analysis on the inverse normal transformation of these residuals. Why TSH concentrations were not directly analysed in the GWAS ? It seems that the GWAs was done differently in Zhou et al and the Est BB study.

The distribution of TSH measurements is non-normally distributed, so analysing untransformed TSH measurements is problematic. All previous GWAS of TSH, including the analysis undertaken by Zhou et al, have applied an inverse normal transformation. This approach also therefore enabled us to meta-analyse our results with those of Zhou et al, giving the largest sample size to date for a GWAS of TSH.

- Line 400 : It should be precised whether the meta-analysis was done using a fixed effect or a random meta-analysis? Has a heterogeneity test between studies been performed?

We have clarified our approach in “Stage 1 analysis” (Methods) and we have added estimates of heterogeneity in our stage 1 analysis to Supplementary Table 2 (column O).

- Line 404 : what was the estimation of the LD Score regression intercept ?

We have added the LD Score regression intercept for the meta-analysis of UK Biobank, EXCEED and Zhou to page 11, paragraph 4 of “Stage 1 analysis” (Methods).

- Line 405 : sentinel SNP should be defined in the text and not only in the legend of tables

We have added a brief description of sentinel variant to the first sentence of “TSH association with 260 variants” (Results), with a fuller description in the Methods.

- Line 423 : « 95% credible set » need to be defined.

This sentence has been expanded to define what a 95% credible set is.

- Lines 450-453 : the definition of hypo-, hyperthyroidism, other thyroid diseases from UKBB data is not clear. These phenotypes are usually tricky to defined and the clinical codes presented in sup Table 15 are not sufficient, the authors should be more explicit on the variables used and how they were combined them to define those diseases. What were the « other thyroid diseases » ? For instance, it is stated in line 146 that secondary hypothyroidism were excluded from the definition of hypothyroidism. This should be detailed in this methodological part.

From my understanding of the paragraph lines 146-152, treated hyperthyroidism were analyzed separately, it is not clear why these cases were not considered as hyperthyroidism directly.

Also descriptive statistics on the number of cases and controls used on the analysis of hypo-, hyperthyroidismn, thyroid cancer and other thyroid disease, as well as their distribution by age, sex. Also, I suggest to put the supplementary table 15 in the excel file with all other supplementary tables for more clarity.

Thank you for highlighting this; we have expanded the description of the phenotype definitions in the Methods (page 13, paragraph 3).

As noted in earlier responses, there is a group of codes which describe “hypothyroidism due to treatment”. This treatment will most commonly be due to hyperthyroidism. However, it is also possible that the treatment was for thyroid cancer or very rarely other reasons. For this reason, we considered it more appropriate to retain as a separate category. We have added a brief clarification to this effect on page 5, paragraph 1.

We have added a new supplementary table, Supplementary Table 1, that gives demographic information for the studies contributing to our analysis. In addition, we have moved Supplementary Table 15 (now Supplementary Table 16) to the supplementary tables document. Thank you for these helpful suggestions.

- Line 455 : I suppose that a logistic regression was used, this should be mentionned. How were the controls defined ? Did you use the same group of controls for each thyroid diseases ?

We have clarified our modelling approach and definition of controls in “Associations with clinical thyroid disease” (Methods).

- Line 458 : at what time was current smoking status or pack-years defined ? What was the proportion of smokers in the analysed samples, and the mean pack-years ?

The previous smoking-adjusted analysis has been removed in favour of a look-up in the latest release from the GSCAN consortium which became available after our original submission. Thus we have no longer used smoking data in our analyses, but for completeness, we have added information on smoking behaviour to the new Supplementary Table 1.

- Line 501 : In the polygenic priority score (PoPS) analysis, why the authors considered window size of $\pm 250\text{kb}$ while in all other analyses in the manuscript a window size of $\pm 500\text{kb}$ was considered?

In their paper on PoPS (<https://doi.org/10.1101/2020.09.08.20190561>), Weeks et al show (Supplementary Table 9) that using PoPS alone is sensitive to the window size, and increasing the window size can reduce the precision of prioritising genes. However, combining PoPS with other criteria can improve performance. We followed the strategy

used by the authors of PoPS in Aragam et al (DOI: 0.1038/s41588-022-01233-6), and prioritized genes selected from a +/-250kb window, or a +/-500kb window if no genes were found from the +/-250kb window.

- In the Supplementary Figures file, after Figure S4, there is a "Supplementary note" that is the same than the Supplementary notes shown before Figure S1 and in the next page there is a Figure that is in the main paper.

We believe this was an issue with the uploading/conversion of the supplementary files and should not appear in the resubmitted supplementary files. Thank you for making us aware.

- Reference #7 (DeepPheWAS) needs to be updated. It is now published in the "Bioinformatics" journal.

- Reference #15 (FinnGen) needs to be updated. It is now published in the "Nature" journal.

Thank you for highlighting these two references. We have updated them accordingly.

REVIEWER COMMENTS

Reviewer #1 (Remarks to the Author):

I would like to thank the Authors for taking into consideration all my comments. It is still not fully clear to me what is the advantage to have split the analysis into stage-1 and stage-2, but I respect the Author's choice and have no further concerns in this regards. I'd like to highlight the following remaining points:

1) The figure on the comparison of $-\log_{10}$ p-values between a censored and uncensored TSH analysis that the Authors provided in response to my question is very informative of the power loss due to censoring. This result will be very informative for future studies in the field and would recommend to include it as a supplementary figure.

2) The very final sentence of Discussion may still give the impression that the developed polygenic scores may predict the age of onset in a single person ("The PGS we developed predicts risk of onset and age of onset of hypothyroidism and hyperthyroidism, of potential utility in future case finding - strategies"). This is far from reality. Analyses show an association of prevalence with age by PGS quantile, but there is no evidence of how accurate a prediction of age at onset might be in the single individual. Please, revise.

Reviewer #2 (Remarks to the Author):

Overall, the authors clarified many points the reviewers raised. However, there are still some major points to clarify. In particular, they added a new study to the analyses which lead to additional questions on the overall study design.

- It seems that the authors misunderstood the comment 1- from Reviewer 1 who suggested to meta-analyze all GWAS in one stage instead of conducting a 2-stage analysis. Moreover, from my understanding, the authors focused on the 260 loci highlighted in stage 1 for the follow-up analyses and not on the 230 loci that were replicated. If results from replication analysis are not usefull, I would suggest, as recommended by reviewer 1, to conduct a joint analysis of all GWAS in one stage.

- The authors added a new study (Genes & Health) in this new version of the manuscript. While all other studies were conducted in individuals of European ancestry, this study was conducted in individuals of South Asian ancestry. In stage 2, the authors conducted a meta-analysis of this study with the EstBB study to replicate findings from stage 1. I would find it more appropriate to show results for each ethnic group separately, and potentially meta-analyse these findings using more appropriate methods than fixed effect inverse variance weighted meta-analysis that do not take into account differences of LD between ethnic groups. If a trans-ethnic analysis was considered, it is not clear why the authors removed all other ethnic group from the UKBB study and did not use for instance individuals of SA ancestry for a meta-analysis with the Gene & Health study.

- Information on the cohort studies are reported in different parts of the manuscript : UKBB and EXCEED are described in the methods part in « Cohort details » as well as in the supplementary methods, the Estonian Biobank is described in stage 2 of the methods section, while Zhou et al and the Genes & Health are poorly described with only a references reported. For the readability of the manuscript, I would suggest to harmonize the description of all studies, for instance by adding a brief description of the Estonian Biobank, Zhou et al, Genes & Health studies in the cohort details in the Methods section, and give some details of all studies in the supplementary methods with a focus on how the TSH levels was defined in each cohort.

- In page 9, the authors stated that the PGS was derived from the full GWAS statistics while in

page 7, the PGS is said to be derived from the stage 1 analysis. This need to be clarified.

- What is the rationale to use imputed genotype data from 10 000 european individuals from the UKBB as LD reference for the fine-mapping analysis (page 12) ? From my understanding, UK population represent about 50% of the stage 1 population. Why the authors did not used LD from 1000 genomes for instance, especially if imputed genotype data from UKBB were derived from 1000 genomes.

- The PGS for TSH, which is derived from stage 1 (that included UKBB data) was tested for association for other thyroid diseases in UKBB. How were the TSH levels associated to thyroid diseases in the UKBB ? If the traits are not independent, then we could expect that the PGS for TSH is also associated to thyroid diseases in the UKBB. It would have been better to test the PGS for TSH in independent studies with information on thyroid diseases. There are some methods proposed to derived the association between a PGS score and an outcome when only summary statistics are available (such as gtx R package), therefore it may be a more robust approach to test the PGS using summary statistics from the largest GWAS for each thyroid diseases, derived from independent samples.

Minor comments :

- Please define "credible set" the first time this term is used, i. e in page 3 (2nd paragraph).

- In page 7, « We then tested PGS associations across ancestries. ». I guess that the authors tested the PGS association in the UKBB, this need to be mentionned here.

REVIEWER COMMENTS

Reviewer #1 (Remarks to the Author):

I would like to thank the Authors for taking into consideration all my comments. It is still not fully clear to me what is the advantage to have split the analysis into stage-1 and stage-2, but I respect the Author's choice and have no further concerns in this regards. I'd like to highlight the following remaining points:

We once again thank both reviewers for offering their time and expertise, and for enabling us to strengthen and improve both the analysis and the manuscript. Please find below our detailed point-by-point response.

1) The figure on the comparison of $-\log_{10}$ p-values between a censored and uncensored TSH analysis that the Authors provided in response to my question is very informative of the power loss due to censoring. This result will be very informative for future studies in the field and would recommend to include it as a supplementary figure.

We have added this as the new Supplementary Figure 10, and we have summarised the findings in "Phenotype" (Methods, page 11, paragraph 2).

2) The very final sentence of Discussion may still give the impression that the developed polygenic scores my predict the age of onset in a single person ("The PGS we developed predicts risk of onset and age of onset of hypothyroidism and hyperthyroidism, of potential utility in future case finding - strategies"). This is far from reality. Analyses show an association of prevalence with age by PGS quantile, but there is no evidence of how accurate a prediction of age at onset might be in the single individual. Please, revise.

Thank you for highlighting this oversight in our previous revision of the paper. We have now removed the reference to prediction here.

Reviewer #2 (Remarks to the Author):

Overall, the authors clarified many points the reviewers raised. However, there are still some major points to clarify. In particular, they added a new study to the analyses which lead to additional questions on the overall study design.

1) It seems that the authors misunderstood the comment 1- from Reviewer 1 who suggested to meta-analyze all GWAS in one stage instead of conducting a 2-stage analysis. Moreover, from my understanding, the authors focused on the 260 loci highlighted in stage 1 for the follow-up analyses and not on the 230 loci that were replicated. If results from replication analysis are not useful, I would suggest, as recommended by reviewer 1, to conduct a joint analysis of all GWAS in one stage.

Reviewer 1 recommended that we adopt the approach of Skol et al, Nat Genet 2006 (PMID: 16415888). Skol et al describe a 2-stage approach where Stage 1 is a GWAS and Stage 2 includes a subset of genetic variants, and both stages are meta-analysed together in a joint analysis. This is the approach we followed. We note that Reviewer 1 has no further concerns about the joint analysis.

2) The authors added a new study (Genes & Health) in this new version of the manuscript. While all other studies were conducted in individuals of European ancestry, this study was conducted in

individuals of South Asian ancestry. In stage 2, the authors conducted a meta-analysis of this study with the EstBB study to replicate findings from stage 1. I would find it more appropriate to show results for each ethnic group separately, and potentially meta-analyse these findings using more appropriate methods than fixed effect inverse variance weighted meta-analysis that do not take into account differences of LD between ethnic groups.

If a trans-ethnic analysis was considered, it is not clear why the authors removed all other ethnic group from the UKBB study and did not use for instance individuals of SA ancestry for a meta-analysis with the Gene & Health study.

Where there is an adequate number of studies of different ancestries our preference would be to use MR-MEGA (PMID: 28911207) for meta-analysis, to quantify the extent to which heterogeneity in allelic effects is attributable to ancestry. Here, we have too few studies for either MR-MEGA or random effects analyses to perform adequately, so our preference is to present the results separately for each ancestry group in addition to presenting the fixed meta-analysis result. We have added to the Discussion an acknowledgement of this limitation. We have clarified Supplementary Table 2 to denote ancestry alongside the study-level summary statistics (Genes & Health being composed of South Asian participants and other cohorts of European ancestry).

In our design, we prioritised showing the relevance of the TSH associations discovered in UK Biobank European ancestry participants to clinical thyroid disease in UK Biobank non-European ancestry subpopulations (i.e. populations independent of discovery to avoid overfitting, as the reviewer highlights below in point 6).

3) Information on the cohort studies are reported in different parts of the manuscript : UKBB and EXCEED are described in the methods part in « Cohort details » as well as in the supplementary methods, the Estonian Biobank is described in stage 2 of the methods section, while Zhou et al and the Genes & Health are poorly described with only a references reported. For the readability of the manuscript, I would suggest to harmonize the description of all studies, for instance by adding a brief description of the Estonian Biobank, Zhou et al, Genes & Health studies in the cohort details in the Methods section, and give some details of all studies in the supplementary methods with a focus on how the TSH levels was defined in each cohort.

Thank you for highlighting this. We have added more information about each study to "Cohort details" (Methods, page 10, paragraphs 1-5). In addition, we have added more information on the TSH phenotype in each study to "Phenotype" (Methods, page 11, paragraph 1), with further detailed information provided in the **Supplementary Note** (page 2-3).

4) In page 9, the authors stated that the PGS was derived from the full GWAS statistics while in page 7, the PGS is said to be derived from the stage 1 analysis. This need to be clarified.

Thank you. We have revised the wording to clarify our meaning, whilst making clear that the PGS was derived from genome-wide association statistics from stage 1 analysis (page 9, paragraph 2).

5) What is the rationale to use imputed genotype data from 10 000 european individuals from the UKBB as LD reference for the fine-mapping analysis (page 12) ? From my understanding, UK population represent about 50% of the stage 1 population. Why the authors did not used LD from 1000 genomes for instance, especially if imputed genotype data from UKBB were derived from 1000 genomes.

Imputed data for UKBB were not derived from 1000 Genomes but from the Haplotype Reference Consortium and a combined UK10K and 1000 Genomes panel; we have now clarified this in the manuscript (under "Stage 1 analysis" in Methods). This led to a coverage of 97 million variants, compared to 35 million for 1000 Genomes, substantially aiding fine mapping through improved coverage. Accounting also for the populations from Zhou et al, 55% of participants in Stage 1 are from the UK. Taken together, the reference panel derived from the HRC and combined UK10K and 1000 Genomes panel offered the best overall fit to the study aims and populations.

6) The PGS for TSH, which is derived from stage 1 (that included UKBB data) was tested for association for other thyroid diseases in UKBB. How were the TSH levels associated to thyroid diseases in the UKBB ? If the traits are not independent, then we could expect that the PGS for TSH is also associated to thyroid diseases in the UKBB. It would have been better to test the PGS for TSH in independent studies with information on thyroid diseases. There are some methods proposed to derive the association between a PGS score and an outcome when only summary statistics are available (such as gtx R package), therefore it may be a more robust approach to test the PGS using summary statistics from the largest GWAS for each thyroid diseases, derived from independent samples.

To address the potential overfitting issue, we have conducted a sensitivity analysis where we tested association with thyroid disease in an independent subset within UKBB, by excluding individuals who were included in the stage 1 analysis (and those who share at least 2nd degree relatedness with individuals included in stage 1). We observed consistent findings which are provided in Supplementary Table 15. Thank you for highlighting this was not clear. We have added some text to clarify that the sensitivity analysis was conducted in independent individuals (page 7, fourth paragraph of "Polygenic score associations" (Results) and page 8, final paragraph of the Results section).

Minor comments :

7) Please define "credible set" the first time this term is used, i. e in page 3 (2nd paragraph).

We have added some text on page 3 to provide a clearer definition of term "credible set" (page 3, second paragraph of "TSH association with 260 sentinel variants" (Results)).

8) In page 7, « We then tested PGS associations across ancestries. ». I guess that the authors tested the PGS association in the UKBB, this need to be mentioned here.

We have modified the text to make it clear that we then tested PGS association with TSH and T4 across ancestries in UKBB (page 7, third paragraph of "Polygenic score associations" (Results)).